# SPARSE DEEP ADDITIVE MODEL WITH INTERACTIONS: ENHANCING INTERPRETABILITY AND PREDICTABILITY

## ABSTRACT

Recent advances in deep learning highlight the need for personalized models that can learn from small or moderate samples, handle high-dimensional features, and remain interpretable. To address this challenge, we propose the Sparse Deep Additive Model with Interactions (SDAMI), a framework that combines sparsity-driven feature selection with deep subnetworks for flexible function approximation. Unlike conventional deep learning models, which often function as black boxes, SDAMI explicitly disentangles main effects and interaction effects to enhance interpretability. At the same time, its deep additive structure achieves higher predictive accuracy than classical additive models. Central to SDAMI is the concept of an Effect Footprint, which assumes that higher-order interactions project marginally onto main effects. Leveraging this principle, SDAMI employs a three-stage strategy to circumvent the search complexity inherent in direct interaction screening: first, identify strong main effects that implicitly carry information about important interactions; second, exploit this information—through structured regularization such as group lasso—to distinguish genuine main effects from interaction effects; third, build subnetwork for identified main effect and interaction. For each selected main effect, SDAMI constructs a dedicated subnetwork, enabling nonlinear function approximation while preserving interpretability and providing a structured foundation for modeling interactions. Extensive simulations and applications with comparisons confirm SDAMI's in reliability analysis, neuroscience, and medical diagnostics further demonstrate SDAMI's versatility in recovering effect structures across diverse scenarios and addressing real-world high-dimensional modeling challenges.

## 1 INTRODUCTION

Deep learning regression now underpins applications across science, engineering, and biomedicine (Cesario et al., 2024; Collins et al., 2024). Yet most architectures are tuned to data-rich regimes with large sample sizes (He et al., 2020). In many emerging settings—especially personalized AI—the reality is the opposite: modest numbers of samples paired with extremely high feature counts. Such small-$n$, large-$k$ problems are increasingly common as measurement technologies extract thousands of variables from limited observations (Jain, 2002; Stefanicka-Wojtas & Kurpas, 2023; Zhou et al., 2015). Our motivating example comes from neuroscience, where we analyze single-cell activity with roughly $n = 500$ observations and over $k = 11,000$ candidate features. This regime creates a basic tension. Classical deep models risk overfitting because the effective sample size per parameter is tiny, while aggressive dimensionality reduction can discard meaningful biological signal. Addressing this trade-off requires models that scale to high dimensions, remain effective in small samples, and preserve interpretability for scientific discovery.

When data are abundant, conventional deep models can achieve high predictive accuracy but typically operate as "black boxes," obscuring how individual variables and their interactions drive predictions (Wang & Lin, 2021). That can suffice for tasks like image classification or speech recognition, but scientific studies need insight into which effects matter and why (Molnar, 2020). In small-$n$, large-$k$ settings, this need becomes even more urgent: limited samples amplify variance and spurious correlations, making it difficult to identify the truly important variables and to characterize

their effects without an interpretable structure (Hastie et al., 2009). These considerations motivate structured architectures that explicitly encode regression effects. By modeling main effects and interactions, one can deliver the estimated component function of the important effects that aid inference, support diagnostics, and tie predictions back to hypotheses, even when data are limited. For example, a concrete illustration comes from modeling visual cortex responses. We illustrate these challenges later on a V1 fMRI dataset, where thousands of Gabor-like features are measured from only a few hundred images, creating a prototypical small-$n$, large-$k$ scenario. Classical sparse additive models treat simple and complex cell terms as independent main effects, offering flexibility but ignoring biologically plausible higher-order associations (Kay et al., 2008; Vu et al., 2008). These limitations motivate us to propose the Sparse Deep Additive Model with Interactions (SDAMI), a structured deep additive framework that preserves interpretability and enhances predictability while enabling the discovery of nonlinear effects and interactions.

The Sparse Deep Additive Model with Interactions is motivated by a new principle introduced in this work: the *Effect Footprint*. The Effect Footprint posits that higher–order interactions typically leave marginal traces in their constituent variables, meaning that even when an interaction cannot be directly estimated—especially in small-$n$, large-$k$ settings—its presence can still be detected through systematic deviations in the corresponding main-effect regressions. This yields statistically efficient pathways for screening and discovering interactions by examining their lower-dimensional component functions. This principle differs fundamentally from strong (or weak) effect heredity (Bien et al., 2013; Lim & Hastie, 2015; Choi et al., 2010), which requires an interaction to be linear and included only when all (or some) associated main effects are already active. This means heredity imposes structural inclusion rules—preventing models that contain pure interactions without main effects—the Effect Footprint provides a *diagnostic criterion*, formalizing how interactions manifest through marginal projections and enabling their detection even when the main effects themselves are weak or negligible. This distinction is particularly important in applications such as our biological imaging study (Section 6), where interaction-driven signals may exist in the absence of strong univariate effects.

SDAMI operationalizes the Effect Footprint within a deep additive architecture by converting marginal signatures into a scalable mechanism for interaction discovery. It implements this idea through a three-stage procedure. In the first stage, the method identifies strong main effects whose marginal signals may implicitly reflect underlying interactions. In the second stage, it employs structured regularization—including group penalties and hierarchical sparsity (Simon et al., 2013; Yuan et al., 2009; Zhao et al., 2009)—to disentangle genuine main effects from interaction contributions and to introduce nonlinear interaction subnetworks only when supported by the data. Lastly, the network is built according to the subnetwork for identified main effect and interaction. For each selected main effect, SDAMI constructs a dedicated subnetwork, allowing flexible nonlinear function approximation while preserving effect-level interpretability. This design achieves a principled balance between flexibility and transparency: the model captures complex nonlinearities using deep subnetworks while maintaining an additive structure that clarifies the roles of individual variables. Extensive simulations demonstrate that SDAMI reliably recovers effect structure across diverse scenarios and avoids both underfitting of main effects and overfitting of interactions.

**Related Work and Differences.** SDAMI addresses a gap not filled by existing models seeking to combine flexibility with effect-level interpretability. Conventional deep neural networks can represent rich interactions implicitly (He et al., 2020), but their entangled architectures obscure variable contributions and rely on post-hoc attribution tools that often assume local linearity, miss nonlinear structure, and become unstable in low-signal or small-$n$ settings (Molnar, 2020). Additive and partially additive neural models improve interpretability through main-effect decompositions, yet typically depend on effect-heredity assumptions—requiring interactions to appear only when associated main effects are present—which restricts their ability to detect pure or higher-order interactions (Agarwal et al., 2021; Vaughan et al., 2018; Yang et al., 2021). Classical sparse additive methods similarly struggle to capture nonlinear interactions (Fan et al., 2011a; Ravikumar et al., 2009; Fan et al., 2011b). Structured sparsity techniques (Yuan & Lin, 2006; Scardapane et al., 2017) and their deep-learning extensions (Wen et al., 2016; Xu et al., 2023; Chang et al., 2021; Enouen & Liu, 2022; Kim et al., 2022) offer principled group or hierarchical selection, but still inherit heredity-type constraints and do not provide a mechanism for disentangling main effects from interactions or allocating model capacity in a way aligned with effect structure. SDAMI offers a unified alternative: it imposes effect footprint with associated effect detection techniques via a extra pipeline from responses

to the input layer so each selected variable receives its own subnetwork, introduces interaction sub-networks only when data exhibit non-negligible effect footprints, and embeds interaction discovery directly into optimization—allowing models composed purely of interactions when supported by evidence—through structured hierarchical penalties (Patel et al., 2020; Shah, 2016). SDAMI thereby integrates and extends three distinct methodological strands: (i) Sparse additive modeling (SpAM), which yields interpretability by enforcing main-effect sparsity but cannot capture nonlinear interactions; (ii) Neural additive frameworks, which increase flexibility by replacing basis expansions with subnetworks but lack a principled mechanism for detecting pure or higher-order interactions; and (iii) Structured sparsity in deep learning, which enables group or hierarchical selection but still treats main effects and interactions jointly and does not disentangle effect-specific representation capacity. SDAMI uniquely combines additive interpretability with deep-network flexibility by using effect footprints to identify influential variables, disentangling main effects from interactions, and introducing interaction subnetworks.

**Our Contribution.** SDAMI introduces a principled framework for discovering nonlinear interactions in deep regression modeling under small-$n$, large-$k$ regimes by combining two key ideas: leveraging the *Effect Footprint* to identify variables likely involved in the nonlinear interactions with nonparametric modeling, and structuring deep subnetworks according to the hierarchy of regression effects. The Effect Footprint enables screening of interaction-relevant variables before exploring the full interaction space, yielding an efficient search in high dimensions. SDAMI then applies structured sparsity to separate main effects from interactions and to introduce interaction subnetworks only when supported by data. Detailed contributions are summarized below:

- Introduce an additive-plus-interaction framework for small-$n$, large-$k$ settings that identifies key main effect components and a single high-dimensional interaction component for constructing the input layer of a deep regression. This recovered interaction component enables reconstruction of pairwise or higher-order interaction components.

- Introduce the *effect footprint* and formalize the marginal-to-interaction connection via Hoeffding-Sobol decomposition, providing a principled mechanism to detect interaction-only variables without assuming heredity constraints—a departure from existing hierarchical sparsity methods.

- Unlike post-hoc interpretation methods, the structured deep additive model with interactions enforces sparsity and interpretability during training via input-layer norm constraints (3), achieving effect level interpretability.

- Explore theorem to illustrate the condition for footprints detectability (4.1), *effect-level selection consistency* under group sparisty (4.2), and *prediction convergence in probability* (4.3)-establishing SDAMI's statistical rigor, and the failure of footprints is rare, impractical edge cases involving perfect independence/symmetry).

- Provide a flexible and interpretable framework that overcomes the pairwise search-space limitation through *effect footprint*, where overcome the $k^2$ searching complexity for second-order criteria. In simulations and applications, we show improved *predictability and interpretability* compared to state-of-the-art interpretable models with high true positive rate (TPR)/low false positive rate (FPR) and informative component-function visualizations.

## 2 PROBLEM SETUP AND RESPONSE-GUIDED STRUCTURED DEEP FRAMEWORK

We observe regression data $\{(\mathbf{X}_i, Y_i)\}_{i=1}^n$, where $\mathbf{X}_i = (X_{i1}, \ldots, X_{ik})^\top \in \mathbb{R}^k$ denotes the predictors and $Y_i \in \mathbb{R}$ is the response. The true regression function is assumed to follow a sparse additive-plus-interaction structure of the form

$$Y_i = \sum_{j \in \mathcal{M}} f_j(X_{ij}) + f(\mathbf{X}_{i,\mathcal{I}}) + \epsilon_i, \tag{1}$$

where $\mathcal{M} \subseteq \{1, \ldots, k\}$ is the index set of important main effects, $\mathcal{I} \subseteq \{1, \ldots, k\}$ is the index set of variables entering the interaction component, and $\epsilon_i$ is a random error with $\mathbb{E}[\epsilon_i] = 0$ and $\mathrm{Var}(\epsilon_i) = \sigma^2$. We assume $|\mathcal{M}| = p \ll k$, so that only a small fraction of predictors directly

contribute as main effects. We define $|\mathcal{I} \setminus \mathcal{M}| = q$, capturing variables that contribute exclusively through interactions but not as main effects. The sets $\mathcal{M}$ and $\mathcal{I}$ are not necessarily nested. In general, $\mathcal{I}$ may contain variables that contribute only through interactions but not as main effects, i.e., $q \neq 0$, including a scenario corresponds to interaction-only effects.

To estimate the model (1), each main-effect function $f_j$ and the interaction function $f_{\mathcal{I}}(\cdot)$ are represented by dedicated neural subnetworks. Let $\theta_j$ and $\theta_{\mathcal{I}}$ denote their respective parameters. Denote by $W_{\mathcal{M},j}^{(1)}$ the weight vector in the first hidden layer connecting input $X_j$ to its main-effect subnetwork, and by $W_{\mathcal{I},j}^{(1)}$ the weight vector connecting $X_j$ to the interaction subnetwork. The estimation problem is then formulated as

$$\min_{\theta} \frac{1}{n} \sum_{i=1}^{n} \left( Y_i - \sum_{j \in \mathcal{M}} \mathrm{NN}^{(j)}(X_{ij}; \theta_j) - \mathrm{NN}^{(\mathcal{I})}(\mathbf{X}_{i,\mathcal{I}}; \theta_{\mathcal{I}}) \right)^2, \tag{2}$$

subject to $\quad \|W_{\mathcal{M},j}^{(1)}\|_{\infty} \leq \kappa_{\mathcal{M}} \|f_j\|, \quad j = 1, \ldots, k, \qquad \|W_{\mathcal{I},j}^{(1)}\|_{\infty} \leq \kappa_{\mathcal{I}} \|f_{\mathcal{I}}\|, \quad j \in \mathcal{I}. \tag{3}$

Here, $\mathrm{NN}^{(j)}(X_{ij}; \theta_j)$ denotes a *neural network (NN) submodule* dedicated to the $j$-th main effect, parameterized by weights $\theta_j$, while $\mathrm{NN}^{(\mathcal{I})}(\mathbf{X}_{i,\mathcal{I}}; \theta_{\mathcal{I}})$ denotes a subnetwork for the interaction set $\mathcal{I}$, parameterized by $\theta_{\mathcal{I}}$. Each NN is a standard feedforward network with hidden layers and nonlinear activations, serving as a flexible nonlinear approximator. The reference functions $f_j$ and $f_{\mathcal{I}}$ represent the true main-effect and interaction-effect components of the regression function $f^{\star}$. The constraints in equation 3 regulate the first-layer weights $W^{(1)}$ relative to $\|f_j\|$ and $\|f_{\mathcal{I}}\|$, ensuring that each subnetwork remains aligned with the magnitude of its corresponding effect and thereby preserving hierarchical structure and interpretability. If $\|f_j\| = 0$, the outgoing weights $W_{\mathcal{M},j}^{(1)}$ vanish, excluding $X_j$ from its subnetwork. Similarly, if $\|f_{\mathcal{I}}\| = 0$, connections into the interaction subnetwork are eliminated. Thus sparsity and interpretability are achieved not through explicit penalties, but through norm-based constraints that prune irrelevant effects, while the loss in equation 2 enforces predictive accuracy.

Direct the optimization of the constrained problem (2) without additional structure becomes infeasible in high dimensions, since it is difficult to distinguish relevant main effects from irrelevant variables or latent contributors to interactions. Another challenge in discovering the interaction component is that even if one restricts attention to pairwise interactions, the number of candidate interaction pairs grows quadratically with the feature count—namely, $\binom{k}{2}$ possible pairs. To overcome these challenge, we introduce the principle of an *effect footprint*, which provides a mechanism for linking variable screening directly to the objective function and guiding the activation of subnetworks in a statistically coherent manner.

## 3 FITTING SPARSE DEEP ADDITIVE MODELS WITH INTERACTIONS (SDAMI)

The constrained optimization problem (2) and (3) present a fundamental challenge in high dimensions: directly solving for all parameters becomes computationally infeasible and statistically unreliable when the number of potential effects far exceeds the sample size. To address this, we propose a principled three-stage procedure (Algorithm A) that leverages the *effect footprint* principle to systematically identify and estimate both main effects and interaction effects: and interaction-candidate indices $\widehat{\mathcal{I}}$ using Group LASSO.

**(i.) Screening via Effect Footprints** Recall the true model 1 from Section 2, the core innovation is recognizing that even if $X_j \notin M$ (no standalone main effect), its participation in the interaction $f(X_{\mathcal{I}})$ can induce a detectable marginal signal.

**Definition 3.1** (Effect Footprint)**.** If $X_j \in \mathcal{I}$, the *effect footprint* is defined as:

$$m_j(x) = \mathbb{E}\big[f(\mathbf{X}_{\mathcal{I}}) \mid X_j = x\big];$$

Although the true model contains no independent term $f_j(X_j)$, the conditional expectation $m_j(x)$ may vary with $x$, thereby creating marginal dependence. This marginal dependence is precisely

what make $X_j$ detectable via univariate screening, despite lacking an independent main effect. By recognizing and leveraging these footprints, we can identify interaction-only variables without exhaustively searching the $\binom{k}{2}$ space of candidate pairs.

To operationalize this principle, we introduce an augmented additive model that captures both true main effects and footprints:

$$Y_i = \sum_{j=1}^{p+q} f_j(X_{ij}) + \epsilon_i, \tag{4}$$

where $\{1, \ldots, p\} = \mathcal{M}$ correspond to true main effects and $\{p+1, \ldots, p+q\} = \mathcal{I} \setminus \mathcal{M}$ represent footprint variables. Let $\mathcal{S} = \{1, \ldots, p+q\}$ denote the union of main and footprint variables. Recovering $\mathcal{S}$ is the first step toward solving the constrained optimization problem (2). Motivated by this augmented model (4), we can apply a sparse additive screening procedure (e.g., sure independence screening for additive models) to identify an estimated active set $\widehat{\mathcal{S}}$ (Ravikumar et al., 2009). This step retains variables with either genuine main effects or non-negligible footprints, while shrinking all others to zero.

**(ii.) Decomposing into Main vs. Interaction Effects**  After screening, the selected features $\widehat{\mathcal{S}}$ are assigned to both main effect $\widehat{\mathcal{M}}$ and interaction sets $\widehat{\mathcal{I}}$, allowing for overlap when a feature has both strong individual predictive signal and meaningful interaction effects. This decomposition is achieved group LASSO with orthogonal basis expansion (Yuan & Lin, 2006). The sets $\widehat{\mathcal{M}}$ and $\widehat{\mathcal{I}}$ are associated with penalty parameters $\lambda_1$ and $\lambda_2$, which are selected via Mallow's $C_p$ Mallows (1973) and cross-validation, respectively.

In this stage, we employ a group LASSO penalty to identify the set of features involved in non-additive interactions. This enables recovery of the regression function as a sum of interaction components corresponding to the supports selected by the group LASSO. Under the effect footprint principle 3.1, and depending on the network architecture, this approach extends beyond pairwise terms to capture higher-order interactions, thereby generalizes heredity-constrained modeling with greater modeling flexibility.

Use the decomposition to construct dedicated subnetworks with norm constraints and solve the constrained optimization problem. The decomposition is justified by the *effect footprint* principle: variables participating exclusively in interactions leave marginal signals detectable via univariate screening, even without standalone main effects.

**(iii.) Solving (2) and (3) via Deep Learning with Norm Constraints**  After $\widehat{\mathcal{M}}$ and $\widehat{\mathcal{I}}$ are determined, these subsets guide the fitting of deep regression model defined in model (1), implemented in PyTorch. Figure 3 illustrates the SDAMI architecture and how structured constraints impose sparsity on the network. The norm-based constraints in the original constrained optimization problem ensure that only the identified main-effect and interaction subnetworks remain active for prediction.

If all variables in $\widehat{\mathcal{I}}$ are fed into a shared interaction subnetwork $\mathrm{NN}^{(\mathcal{I})}(\mathbf{X}_{i,\mathcal{I}}; \theta_{\mathcal{I}})$, which learns the joint interaction structure is denoted as SDAMI. Otherwise, if each pairwise interaction $(x_i, x_j) \in \widehat{\mathcal{I}}$ is fed to a separate subnetwork that learns individual interaction structures, we denote it as SDAMI$-p$. While SDAMI$-p$ focuses on pairwise interactions for computational tractability (reducing the search from $\binom{k}{2}$ to only relevant pairs), our higher-order variant (SDAMI) captures arbitrary-order feature interactions. Unlike previous approaches (Enouen & Liu, 2022; Kim et al., 2022), SDAMI leverages effect-footprint-based screening to systematically reduce the feature set prior to modeling any-order interactions, avoiding the combinatorial explosion of the unapproachable $\binom{k}{2}$ space. Although the two variants employ different architectures, the training algorithm remains identical. A detailed algorithmic description is provided in Appendix A of the supplementary material. Neural network architectures (depth, width) are selected via 5-fold cross-validation on a held-out validation set. Detailed configurations for each dataset are provided in Appendix E.

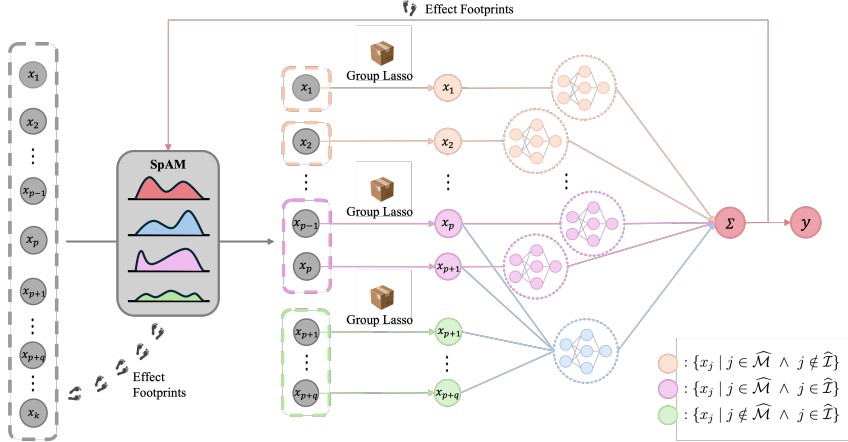

Figure 1: The SDAMI architecture. Screening identifies both main and footprint variables, which guide the activation of subnetworks and enforce biologically and statistically meaningful structure.

## 4 THEORETICAL ANALYSIS: THE ROLE OF EFFECT FOOTPRINT, SELECTION CONSISTENCY, MODEL CONVERGENCE

Under the model space definition mentioned in Appendix B, we present the theoretical foundation of SDAMI in three parts. First, we formalize the concept of *effect footprint*, which justifies feasible high-dimensional screening. Second, we show that SDAMI attains *effect-level selection consistency*, recovering both the true main effects and the interaction structure. Finally, we establish predictive validity by proving that the fitted predictor converges in probability to the true model (1).

**Theorem 4.1** (When effect footprints vanish). *Let* $\mathbf{X}_\mathcal{I} = (X_j, \mathbf{Z})$ *be the variables in an interaction* $f(\mathbf{X}_\mathcal{I})$ *with* $\mathbb{E}[f(\mathbf{X}_\mathcal{I})] = 0$. *Define*

$$m_j(x) = \mathbb{E}[f(\mathbf{X}_\mathcal{I}) \mid X_j = x].$$

*Then* $m_j(x)$ *is constant (no footprint) iff the first–order projection of* $f$ *onto functions of* $X_j$ *vanishes in the Hoeffding–Sobol decomposition (Sobol', 1990; Sobol, 2001). In this case,* $f$ *contains only higher–order components involving* $X_j$.

This characterization isolates the exceptional cases in which footprints fail: a variable leaves no detectable footprint precisely when its influence appears solely through higher–order interactions that vanish after averaging over the remaining inputs. Such a variable may still be essential via interactions, but univariate screening cannot detect it. Two canonical settings illustrate this: (i) independence with centering (e.g., bilinear forms of independent, mean-centered inputs), and (ii) perfect symmetry with antisymmetric interactions (e.g., the XOR rule for binary data or odd functions under symmetric continuous inputs). These conditions are stringent; in practice predictors are correlated, distributions seldom perfectly symmetric, and noise disrupts exact cancellations. Consequently, footprints typically exist, providing a robust signal for screening. A detailed proof is given in Appendix B of the supplementary material.

**Theorem 4.2** (Effect-level selection consistency of SDAMI). *Under assumptions (A1)–(A7),*

$$\mathbb{P}\Big(\{j : \widehat{f}_j \neq 0\} = \mathcal{M} \text{ and } (\widehat{f}_\mathcal{I} \neq 0 \Leftrightarrow f_\mathcal{I} \neq 0)\Big) \longrightarrow 1 \quad as\ n \to \infty.$$

Thus SDAMI does not merely exploit footprints heuristically; it achieves a rigorous form of oracle recovery. As $n$ grows, SDAMI selects exactly the true set of main effects and correctly detects the interaction with probability tending to one, ensuring that the discovered structure reflects the underlying generative mechanism. The proof (Appendix C of the supplementary material) employs a block-wise primal–dual witness argument for the group-lasso formulation, leveraging footprint-induced group signals and oracle inequalities for group sparsity (Lounici et al., 2011; Negahban et al., 2009).

| Case | Functional Form | Conceptual Description |
|------|-----------------|------------------------|
| 1 | $y = f_1(x_1) + f_2(x_2) + f_3(x_3) + f_4(x_4)$ | Only strong main effects, no interactions |
| 2 | $y = f_1(x_1) + f_2(x_2) + f_3(x_3) + 0.01 f_4(x_4)$ | Main effects with weak signals |
| 3 | $y = f_1(x_1) + f_2(x_2) + f_3(x_3) + f_5(x_4, x_5)$ | Main effects plus one interaction block with no overlap |
| 4 | $y = f_1(x_1) + f_2(x_2) + f_3(x_3) + f_5(x_3, x_4)$ | Main effects + 1 interaction block with some overlapping variables |
| 5 | $y = f_1(x_1) + f_2(x_2) + f_3(x_3) + f_5(x_2, x_3)$ | Main effects plus one interaction block with all variables overlapping |
| 6 | $y = f_5(x_1, x_2) + f_5(x_3, x_4)$ | Only interaction effects, no main effects |

Table 1: The summary table for numerical simulation models.

**Theorem 4.3** (Prediction convergence in probability for SDAMI). *Let $\widehat{A}_n$ be the SDAMI-selected index set and let $\widehat{f}_n$ be the SDAMI estimator. Suppose* (B1)–(B6) *hold. Then, for every fixed $\varepsilon > 0$,*

$$\mathbb{P}\Big( |\widehat{f}_n(\mathbf{X}) - f^\star(\mathbf{X})| \geq \varepsilon \Big) \longrightarrow 0 \quad \text{as } n \to \infty,$$

**Remark 4.4.** Our convergence result Theorem 4.3 relies on Assumption (B4), which requires that the trained network achieves near-optimal empirical risk. This is a standard assumption in deep learning theory and is empirically validated by our experiments. Recent advanced in optimization theory for over-parameterized networks provide conditions under which gradient descent provably converges to global minima, leading support to this assumption

The key idea is to combine sieve approximation with uniform generalization. Selection consistency concentrates learning on the correct coordinates; empirical risk minimization up to a vanishing tolerance, together with a uniform law of large numbers for squared loss (via Rademacher and covering bounds for norm–constrained networks), transfers empirical to population $L_2$–risk (Bartlett & Mendelson, 2002; Mohri et al., 2018; van de Geer, 2000). In parallel, ReLU approximation theory ensures the sieve approximates the oracle regression under a suitable growth schedule (Barron, 1993; Yarotsky, 2017; Schmidt-Hieber, 2020; Suzuki, 2019). A uniform $L_2$ envelope (implied by norm constraints and square-integrability) guarantees uniform integrability, so vanishing population risk implies vanishing misfit probability via a Markov-type bound. Full details appear in Appendix D of the supplementary material.

## 5 NUMERICAL EXPERIMENTS

We conduct comprehensive numerical simulations to evaluate SDAMI's ability to recover effect structures and achieve predictive accuracy across diverse scenarios. Data are generated under six distinct settings (Table 1), each defined by different functional forms involving main effects, interactions, or both, with varying degrees of variable overlap.

For each setting, (1) Sample sizes $n$ vary across $150, 300$, and $450$; (2) Feature dimension is fixed at $k = 150$ in a high-dimensional regime, with only a few features having substantive effects; (3) Responses are generated as $Y_i = \sum_{j \in \mathcal{M}} f_j(X_{ij}) + f(X_{i,\mathcal{I}}) + \epsilon_i$, where $X_i \sim \text{Uniform}(-2.5, 2.5)$ independently and $\epsilon_i \sim N(0, \sigma^2)$; (4) True functions are drawn from representative nonlinear forms: $f_1(x) = -2\sin(2x)$, $f_2(x) = \frac{x^2}{2} + 1$, $f_3(x) = x - \frac{1}{2}$, $f_4(x) = e^{-x} + e^{-1} - 1$, and $f_5(x_1, x_2) = e^{\sin(x_1) + \cos(x_2) - 1}$. Detailed formulations for the six experimental cases are provided in Table 1.

We benchmark SDAMI and SDAMI$-p$ against state-of-the-art interpretable models including neural additive model (NAM) which treats interaction ad hoc via heredity (Agarwal et al., 2021), GAMI-Net which uses soft hierarchical constraints limiting pure interactions (Yang et al., 2021), NODE-GAM, and NODE-GA$^2$M (Chang et al., 2021) which applies heredity through structured penalties, as well as deep neural networks (DNN), fast sparse additive models (fSpAM), and LASSO. Architecture selection for SDAMI a d SDAMI$-p$ is determined by cross-validation. The hyperparameters, optimal network architectures, and computing times are summarized in Appendix E, Appendix F, and Appendix H, respectively.

Across all simulation settings and sample sizes, SDAMI consistently achieves the lowest mean squared error (MSE) (Tables 2), confirming its capacity to flexibly capture nonlinear main and interaction effects while maintaining interpretability. In Case 1, which involves only strong main effects, SDAMI attains the best accuracy without introducing spurious interactions, demonstrating

|  | SDAMI | DNN | fSpAM | LASSO | NAM | GAMI-NET | NODE-GA$^2$M | NODE-GAM |
|--------|--------|--------|--------|--------|--------|----------|-------------|----------|
| Case 1 | $\mathbf{0.68}_{\pm 0.59}$ | $14.37_{\pm 1.03}$ | $5.84_{\pm 0.52}$ | $4.77_{\pm 0.94}$ | $14.36_{\pm 1.35}$ | $5.83_{\pm 3.09}$ | $1.31_{\pm 1.53}$ | $2.42_{\pm 2.52}$ |
| Case 2 | $\mathbf{0.57}_{\pm 0.75}$ | $5.39_{\pm 0.42}$ | $3.22_{\pm 0.25}$ | $3.02_{\pm 0.37}$ | $5.19_{\pm 0.42}$ | $2.26_{\pm 1.05}$ | $0.72_{\pm 0.81}$ | $1.58_{\pm 1.64}$ |
| Case 3 | $\mathbf{0.58}_{\pm 0.88}$ | $5.78_{\pm 0.43}$ | $3.55_{\pm 0.25}$ | $3.61_{\pm 0.39}$ | $5.68_{\pm 0.49}$ | $2.56_{\pm 1.02}$ | $1.00_{\pm 1.10}$ | $1.83_{\pm 1.95}$ |
| Case 4 | $\mathbf{0.52}_{\pm 0.91}$ | $7.11_{\pm 0.51}$ | $3.53_{\pm 0.27}$ | $3.34_{\pm 0.40}$ | $6.95_{\pm 0.56}$ | $3.05_{\pm 1.29}$ | $0.96_{\pm 1.07}$ | $1.82_{\pm 1.93}$ |
| Case 5 | $\mathbf{0.44}_{\pm 0.79}$ | $5.90_{\pm 0.43}$ | $3.65_{\pm 0.27}$ | $3.59_{\pm 0.41}$ | $5.77_{\pm 0.48}$ | $2.33_{\pm 1.18}$ | $0.85_{\pm 0.99}$ | $1.75_{\pm 1.83}$ |
| Case 6 | $\mathbf{0.46}_{\pm 0.24}$ | $1.05_{\pm 0.11}$ | $0.63_{\pm 0.05}$ | $0.61_{\pm 0.10}$ | $0.97_{\pm 0.09}$ | $5.85_{\pm 3.09}$ | $0.48_{\pm 0.29}$ | $0.52_{\pm 0.33}$ |

Table 2: The performance for 6 different case type when $n = 150$. We show the average root mean squared error (RMSE) over 100 simulations $\pm$ the standard deviation.

| Method | SDAMI | | LASSONET | | SODA | |
|--------|-------|-----|----------|-----|------|-----|
|  | TPR ↑ | FPR ↓ | TPR ↑ | FPR ↓ | TPR ↑ | FPR ↓ |
| Case 1 | $\mathbf{1.0000}\ (-)$ | $1.1 \times 10^{-5}\ (-)$ | 0.4900 (0.0490) | 0.0037 (0.0028) | 0.0175 (0.0641) | $6 \times 10^{-4}\ (2 \times 10^{-4})$ |
| Case 2 | $\mathbf{1.0000}\ (-)$ | $1.1 \times 10^{-5}\ (-)$ | 0.2550 (0.0350) | 0.0099 (0.0138) | 0.0475 (0.1048) | $1 \times 10^{-3}\ (2 \times 10^{-4})$ |
| Case 3 | $\mathbf{0.7500}\ (-)$ | $10^{-4}\ (10^{-5})$ | 0.1400 (0.3007) | 0.1724 (0.2810) | 0.025 (0.0754) | $6 \times 10^{-4}\ (3 \times 10^{-4})$ |
| Case 4 | $\mathbf{0.7600}\ (-)$ | $10^{-4}\ (10^{-5})$ | 0.1300 (0.3051) | 0.1621 (0.2701) | 0.040 (0.1049) | $7 \times 10^{-4}\ (3 \times 10^{-4})$ |
| Case 5 | $\mathbf{0.7550}$ (0.0249) | $10^{-4}\ (10^{-5})$ | 0.1250 (0.2947) | 0.1629 (0.2814) | 0.055 (0.1100) | $9 \times 10^{-4}\ (2 \times 10^{-4})$ |
| Case 6 | $\mathbf{0.6000}\ (-)$ | $10^{-4}\ (10^{-5})$ | 0.1100 (0.2700) | 0.1432 (0.2334) | - (-) | $6 \times 10^{-4}\ (2 \times 10^{-4})$ |

Table 3: Mean and standard deviation of TPR (Higher is better) and FPR (Lower is better) over 100 simulations from SDAMI, LASSONET, SODA when $n = 150$ where $(-)$ indicates value $< 1e^{-5}$.

parsimony (Tables 2). In Case 2 (weak signals), SDAMI continues to outperform benchmarks, reflecting robustness to small effect sizes. In Cases 3–5, which include both main and interaction effects with varying degrees of overlap, fSpAM, LASSO, NAM, and GAMI-NET show limited capacity to recover the true structures, while SDAMI consistently models both overlapping and non-overlapping interactions, achieving markedly lower errors across all sample sizes. In Case 6, where effects arise solely from interactions, SDAMI retains strong predictive performance, while others deteriorate substantially and DNN suffers from instability. Taken together, the results across Tables 2 demonstrate that SDAMI provides a balanced combination of flexibility, interpretability, and accuracy. By leveraging effect footprints, it adapts to diverse data-generating mechanisms and consistently outperforms existing approaches, validating its utility as a powerful framework for sparse high-dimensional regression in the presence of complex effect structures. The additional numerical experiments with respect to different sample size is displayed in Appendix G of the supplementary material.

We further evaluate the feature selection performance of SDAMI, focusing on its ability to recover true main and interaction effects while minimizing false discoveries. TPR measures the proportion of truly relevant features (main effects and interactions) that are correctly identified and FPR measures the proportion of irrelevant features incorrectly identified as relevant. The total searching space is $\binom{k}{2}$ because of the combinations of number of feature up to second order. This provides a common benchmark for the combinatorial difficulty of interaction discovery; however, different algorithm may or may not explicitly form all such terms. SDAMI uses effect-footprint screening to reduce the set of candidate variables, thereby substantially shrinking the effective searching space compared to a naive $\binom{k}{2}$ expansion. Table 3 summarizes the TPR and FPR when $n = 150$, averaged over 100 simulations and compared with LASSONET(Lemhadri et al., 2021), and sodavis (SODA)(Li, 2015). TPR measures the proportion of correctly identified active variables, while FPR reflects the rate of spurious selections. SDAMI achieves near-perfect TPRs of 1.0 in Cases 1 and 2, dominated by main effects, showing it reliably identifies relevant signals without omission. In more complex settings with overlapping and non-overlapping interactions (Cases 3–6), SDAMI maintains substantially higher TPRs than LASSONET and SODA, which experience steep sensitivity drops. Concurrently, SDAMI obtains extremely low FPRs, often on the order of $10^{-4}$, whereas competitors select irrelevant features at much higher rates. This result indicates that SDAMI strikes a favorable balance between sensitivity and specificity, crucial for high-dimensional regression where false discoveries can obscure interpretation. Stability across 100 replications affirms robustness , while improvements from $n = 150$ to $n = 300$ confirm scalability. Overall, SDAMI demonstrates reliable, precise feature recovery in sparse, high-dimensional problems with complex effect structures. The additional experiment of feature selection for $n = 300$ is shown in Appendix G of the supplementary material.

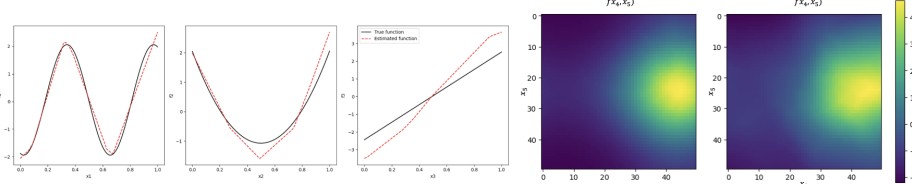

Figure 2: (Case 3) The three figures on the left: Estimated (red dashed lines) versus true additive component functions (solid black lines) for three main effects; the two figures on the far right: the first shows the true response surface for interaction, and the second shows its estimated response surface.

In our simulation studies, a key advantage of SDAMI over other machine learning models is its interpretability through visualization. Figure 2 illustrates Case 3 results, where the black solid line shows the true function and the red dashed line shows SDAMI estimates, demonstrating accurate recovery of complex nonlinear patterns. Additionally, visualizations for all simulation cases are provided in Appendix G.2 of the supplementary material, underscoring SDAMI's value for interpretable modeling in high-dimensional regression.

## 6 REVISIT REAL DATASETS FOR BETTER UNDERSTANDING PRACTICAL USE OF SDAMI

The V1 fMRI dataset (Kay et al., 2008) records voxel responses from human primary visual cortex at $2\,\mathrm{mm}\times2\,\mathrm{mm}\times2.5\,\mathrm{mm}$ resolution on a 4 T scanner while subjects viewed grayscale natural images through a circular aperture. Stimuli are flashed three times per second with interleaved blanks, and signals are preprocessed to reduce noise and nonstationarity. Prior work shows interaction effects among complex cells (Kay et al., 2008; Vu et al., 2008), but how to model such interactions while preserving biological meaning remains underexplored. To foreground the neuroimaging challenge—small $n$, high $k$—experiments use 300 unique natural images, each summarized by 1,800 Gabor-filter features derived from complex-cell processing; each voxel reflects pooled, rectified activity organized by a receptive-field hierarchy over space, frequency, and phase. The pipeline of generating V1-cell response is summarized in Appendix I.3. Figure 11 sketches the pipeline producing simple-cell and complex-cell predictors (and Figure 12 shows the SDAMI linkage to voxel responses).

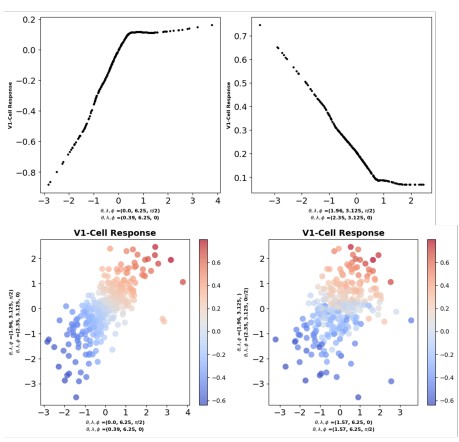

Figure 3: (V1 Cell Dataset) Upper panel: the predicted Marginal main effects; lower panel: the estimated response surface for interactions.
i

Across the seven real-world datasets in Table 4, SDAMI achieves the best or near-best performance in 5 out of 7 cases, with particularly strong gains on Chip, V1 Cell, and BikeShare. While NODE-GA$^2$M shows marginal advantages on California Housing and competitive performance on Wine, SDAMI's systematic superiority across diverse application domains—demonstrates its robustness. Importantly, SDAMI achieves these improvements while maintaining effect-level interpretability through component visualization, providing both predictive accuracy and scientific transparency.

The interaction subnetworks prove particularly valuable in domains with known higher-order dependencies, such as neuroscience where complex-cell receptive fields arise from nonlinear pooling of simple-cell quadrature pairs. Figure 3 displays estimated main effects from selected Gabor-filter features (highlighting positions, orientations, and scales linked to activity) and interaction surfaces

|  | SDAMI | DNN | fSpAM | LASSO | LASSONET | NAM | GAMI-NET | NODE-GAM | NODE-GA$^2$M | Scale |
|---|---|---|---|---|---|---|---|---|---|---|
| Chip | **0.244** | 0.927 | 0.753 | 0.276 | 0.904 | 0.967 | 0.495 | 0.455 | 0.546 | ×1 |
| Diabetes | **0.524** | 0.584 | 0.588 | 0.595 | 0.566 | 0.556 | 0.542 | 0.622 | 0.674 | ×0.01 |
| V1 Cell | **0.622** | 0.702 | 0.793 | 0.789 | 0.792 | − | 0.734 | 0.772 | 0.739 | ×1 |
| Wine | **0.672** | 0.702 | 0.771 | 0.745 | 0.703 | 0.712 | 0.701 | 0.721 | 0.698 | ×1 |
| BikeShare | **0.440** | 0.459 | 1.484 | 1.434 | 0.468 | 1.001 | 0.592 | 1.001 | 0.554 | ×0.01 |
| CA Housing | 0.529 | 0.531 | 0.821 | 0.731 | 0.536 | 0.579 | 0.528 | 0.571 | **0.503** | ×1 |

Table 4: The performance for 7 medium-sized datasets. All of them are regression datasets and shown the Root Mean Squared Error (RMSE). $(-)$ indicates model is infeasible for the dataset.

for key feature pairs, revealing synergistic patterns consistent with cortical pooling. The primary visual cortex (V1) processes visual information through a hierarchical organization where simple cells respond to oriented edges at specific spatial positions and frequencies, characterized by Gabor filter parameters including orientation angle $\theta$, spatial wavelength $\lambda$, and phase $\phi$. The upper left and right panel shows complete different trend. The right panel shows the decrease from initial response to near-zero, suggesting this complex cell is driven by orthogonal orientations that suppress its baseline activity. As for the lower two panels visualize pairwise complex cell interactions, where both axes represent the response magnitudes of two distinct complex cell. Both demonstrate the synergistic excitation that the strongest V1 responses occur when both complex cells are simultaneously active, indicating the neuron functions as a feature-conjunction detector responsive to specific spatial configuration such as the corners or interesting edges. The biologic meaning behind the scene is that the simple-cell inputs are integrated nonlinearly that classical additive models cannot capture. SDAMI's ability to automatically discover and visualize such high-order interactions in the high-dimensional Gabor-filter feature space, rather than ad hoc combinations, validates its utility for neuroscientific inference and demonstrate superior predictive accuracy.

Together, these results show that SDAMI delivers competitive or superior prediction and biologically grounded interpretability in small-$n$, large-$k$ regimes, establishing a principled framework for response modeling in neuroscience and other high-dimensional domains. Due to the page limitation, the details of other dataset analyses are given in Appendix I.

# 7 CONCLUSION

This paper introduced the Sparse Deep Additive Model with Interactions, a structured deep learning framework tailored for small-$n$, large-$k$ regression problems. By leveraging the principle of *effect footprints*, SDAMI offers a systematic approach to detecting and modeling higher-order interactions while retaining effect-level interpretability. The method enforces sparsity through norm-based constraints that prune irrelevant variables and subnetworks, ensuring both statistical stability and interpretability. Theoretical analysis established effect-level selection consistency and prediction convergence in probability, providing rigorous guarantees beyond heuristic interpretability. Simulation studies demonstrated that SDAMI reliably recovers both main and interaction effects, outperforming classical additive models and black-box neural networks. Applications to neuroscience and reliability analysis further illustrated the model's versatility and its ability to bridge deep learning with domain-specific interpretability requirements.

**Limitations and Future Directions.**

While SDAMI offers interpretability with statistical guarantees, several limitations remain. First, the three-stage fitting procedure relies on estimating function norms via SpAM, which can be computationally demanding; SIS-based screening (Fan & Lv, 2008; Fan et al., 2011a) could improve scalability. Second, establishing finite-sample convergence rates for both main effects and interaction terms; recent theory in sparse high-dimensional additive models (Gregory et al., 2021) establishes finite-sample bound showing oracle equivalence under sparsity conditions, which may be adapted to prove minimax rates for SDAMI's multi-stage estimators. Third, extending SDAMI with safe screening (Nakagawa et al., 2016) could further reduce computational cost by retaining only necessary interaction candidates. Recent Deep P-Spline work (Hung et al., 2025) shows penalized spline activations provide both statistical efficiency and speed; incorporating such activations into SDAMI alongside convergence theory would deepen understanding of finite-sample behavior and broaden applicability. These directions promise stronger computational efficiency and theoretical rigor.

## REPRODUCIBILITY STATEMENT

We release the source code and configuration files for the main experiments, along with detailed instructions for data generation and model training. All theoretical results are presented in the appendix, accompanied by explanations and underlying assumptions. The implementation has been carefully validated, and empirical results further confirm the correctness of the proposed Sparse Deep Additive Model with Interaction (SDAMI).

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

# SUPPLEMENTARY MATERIAL FOR SPARSE DEEP ADDITIVE MODEL WITH INTERACTIONS: ENHANCING INTERPRETABILITY AND PREDICTABILITY

## A    SDAMI ALGORITHM

This section describes the detail of the SDAMI algorithm and how the model fitting works.

---

**Algorithm 1** SDAMI Fitting

---

**Require:** Data $\{(X_i, Y_i)\}_{i=1}^n$, tuning parameters $\lambda_1, \lambda_2$

1: **Step 1: Effect Footprint Screening (SpAM).**
   - Fit the sparse additive model

$$Y_i = \sum_{j=1}^{p+q} f_j(X_{ij}) + \epsilon_i$$

   using SpAM with penalty $\lambda_1$.
   - Obtain estimated active set $\widehat{\mathcal{S}} \subseteq \{1, \ldots, p+q\}$ containing both true main effects and footprint variables.

2: **Step 2: Decomposition of Active Set (Group Lasso).**
   - Apply group lasso with orthogonal basis expansion on $\widehat{\mathcal{S}}$.
   - Decomposition of into $\widehat{\mathcal{M}}$ and $\widehat{\mathcal{I}}$.
   - Select penalty $\lambda_2$ via cross-validation (with $\lambda_1$ selected by Mallow's $C_p$).

3: **Step 3: SDAMI Model Fitting.**
   - Fit the constrained deep regression model using $\widehat{\mathcal{M}}$ and $\widehat{\mathcal{I}}$.
   - Implement subnetworks in PyTorch, with sparsity imposed via norm-based constraints.

**Ensure:** Estimated main-effect subnetworks $\{\text{NN}^{(j)}\}_{j \in \widehat{\mathcal{M}}}$ and interaction subnetworks $\{\text{NN}^{(\mathcal{I})}\}_{\mathcal{I} \in \widehat{\mathcal{I}}}$.

---

**Regularization Parameter Selection.** The regularization parameters $\lambda_1, \lambda_2$ are selected by minimizing the estimated risk and by cross-validation, respectively. The effective degree of freedom is defined as $\text{df}(\lambda) = \sum_j \nu_j I(\|\hat{f}_j\| \neq 0)$, where $\nu_i = \text{trace}(S_j)$ and $S_j$ denotes the smoothing matrix for the $j$-th dimension. The estimate is given by

$$C_p = \frac{1}{n} \sum_{i=1}^n \left( Y_i - \sum_{j=1}^p \hat{f}_j(X_j) \right)^2 + \frac{2\hat{\sigma}^2}{n} \text{df}(\lambda).$$

## B    PROOF OF THEOREM 4.1

**Model space.** Let $\mathcal{M} \subseteq \{1, \ldots, p\}$ be the index set for additive (univariate) components, and let $\mathcal{I} \subseteq \{1, \ldots, q\}$ be the index set for the multivariate interaction component. For $j \in \mathcal{M}$, let $\text{NN}^{(j)}(x_j; \theta_j)$ denote the univariate neural network, and let $\text{NN}^{(\mathcal{I})}(x_{\mathcal{I}}; \theta_{\mathcal{I}})$ denote the multivariate neural network on coordinates $\mathcal{I}$.

We define the hypothesis classes induced by the first-layer constraints

$$\|W_{\mathcal{M},j}^{(1)}(\theta_j)\|_\infty \leq \kappa_{\mathcal{M}} \|f_j\|, \quad j \in \mathcal{M},$$

$$\|W_{\mathcal{I}}^{(1)}(\theta_{\mathcal{I}})\|_\infty \leq \kappa_{\mathcal{I}} \|f_{\mathcal{I}}\|.$$

The admissible univariate and multivariate function classes are

$$\mathcal{H}_j = \left\{ f_j(\cdot) = \mathrm{NN}^{(j)}(\cdot; \theta_j) : \|W_{\mathcal{M},j}^{(1)}(\theta_j)\|_\infty \leq \kappa_{\mathcal{M}} \|f_j\| \right\}, \qquad j \in \mathcal{M},$$

$$\mathcal{G}_{\mathcal{I}} = \left\{ g(\cdot) = \mathrm{NN}^{(\mathcal{I})}(\cdot; \theta_{\mathcal{I}}) : \|W_{\mathcal{I}}^{(1)}(\theta_{\mathcal{I}})\|_\infty \leq \kappa_{\mathcal{I}} \|g\| \right\}.$$

**Definition B.1** (Model space of the structured neural network). The functional model space associated with the neural network estimator in equation 2–equation 3 is

$$\mathcal{F}_{\mathrm{NN}}(\mathcal{M}, \mathcal{I}) = \left\{ f(x) = \sum_{j \in \mathcal{M}} f_j(x_j) \; + \; g(x_{\mathcal{I}}) \; : \; f_j \in \mathcal{H}_j, \; g \in \mathcal{G}_{\mathcal{I}} \right\}.$$

If sparsity over the sets $\mathcal{M}$ and $\mathcal{I}$ is desired, the overall sparse model space is

$$\mathcal{F}_{\mathrm{NN}}(s_1, s_2) = \bigcup_{\substack{\mathcal{M} \subseteq \{1,\ldots,p\}, \; |\mathcal{M}| \leq s_1 \\ \mathcal{I} \subseteq \{1,\ldots,q\}, \; |\mathcal{I}| \leq s_2}} \mathcal{F}_{\mathrm{NN}}(\mathcal{M}, \mathcal{I}).$$

This section provides the detailed proof of Theorem 4.1, which establishes the equivalence between vanishing effect footprints and the disappearance of the first–order projection in the Hoeffding–Sobol decomposition. The result clarifies when a variable contributes only through higher–order interactions and thus leaves no detectable marginal footprint.

We begin with the Hoeffding–Sobol decomposition. Let $f(\mathbf{X}_{\mathcal{I}})$ be a centered function, i.e., $\mathbb{E}[f(\mathbf{X}_{\mathcal{I}})] = 0$. Then $f$ admits the unique expansion

$$f(\mathbf{X}_{\mathcal{I}}) = f_{\{j\}}(X_j) + \sum_{S \subseteq \mathcal{I}, \, j \in S, \, |S| \geq 2} f_S(\mathbf{X}_S) + \sum_{S \subseteq \mathcal{I}, \, j \notin S, \, |S| \geq 1} f_S(\mathbf{X}_S),$$

where the components $f_S$ are mutually orthogonal in $L^2$, each has mean zero, and $f_{\{j\}}(X_j)$ represents the unique first–order contribution of $X_j$. The remaining terms correspond either to higher–order interactions involving $X_j$ or to effects of variables not involving $X_j$.

Conditional expectation with respect to $X_j$ is the orthogonal projection of $f$ onto the subspace of $L^2$ functions of $X_j$, as ensured by the Doob–Dynkin lemma and the Hilbert projection theorem. Hence the footprint $m_j(X_j) = \mathbb{E}[f(\mathbf{X}_{\mathcal{I}}) \mid X_j]$ coincides with this projection. By uniqueness of the Hoeffding–Sobol components, this projection is exactly $f_{\{j\}}(X_j)$. The two directions now follow. If $f_{\{j\}}$ vanishes identically, then conditioning the decomposition on $X_j$ eliminates all other terms: for $S$ not containing $j$, centeredness of $f_S$ implies $\mathbb{E}[f_S(\mathbf{X}_S) \mid X_j] = 0$, while for $S$ containing $j$ with $|S| \geq 2$, orthogonality ensures $\mathbb{E}[f_S(\mathbf{X}_S) \mid X_j] = 0$. Thus $m_j(X_j) = 0$, which is constant, so $X_j$ leaves no footprint. Conversely, if $m_j(X_j)$ is constant almost surely, then $\mathbb{E}[f(\mathbf{X}_{\mathcal{I}}) \mid X_j]$ is identically zero because $f$ is centered. Since this conditional expectation is the projection of $f$ onto the space of functions of $X_j$, it follows that $f_{\{j\}}(X_j) \equiv 0$.

Therefore, the footprint $m_j(x)$ is constant if and only if the first–order projection $f_{\{j\}}(X_j)$ vanishes. In this case, the variable $X_j$ contributes only through higher–order interactions, and its marginal influence disappears in expectation, thereby proving Theorem 4.1.

## C  CONDITIONS AND PROOF OF THEOREM 4.2

This section establishes the effect-level selection consistency of SDAMI. We begin by introducing the technical assumptions that govern the noise, design structure, signal strength, and basis expansion. These conditions provide the foundation for analyzing the group-lasso estimator used in SDAMI and for verifying the primal–dual witness construction that guarantees selection consistency.

**Assumption C.1** (Conditions for effect-level selection).

(A1) (*Noise*) The errors $\epsilon_i$ in the true function (1) of the main paper are sub-Gaussian with mean zero and variance proxy $\sigma^2$.

(A2) (*Within-group orthonormality*) For each main effect $j$,

$$\frac{1}{n}\Phi_j^\top \Phi_j = I,$$

and for the interaction block $\Phi_\mathcal{I}$,

$$\frac{1}{n}\Phi_\mathcal{I}^\top \Phi_\mathcal{I} = I, \qquad \frac{1}{n}\Phi_\mathcal{I}^\top \Phi_j = 0 \;\; (j \in \mathcal{I}).$$

(A3) (*Block coherence*) For $g \neq g'$,

$$\left\| \tfrac{1}{n} X_g^\top X_{g'} \right\|_{\mathrm{op}} \leq \mu < 1,$$

where $X_g$ denotes the block of design columns for group $g$.

(A4) (*Restricted eigenvalue*) The Gram matrix on the active set

$$\Sigma_{A^\star A^\star} = \tfrac{1}{n} X_{A^\star}^\top X_{A^\star}, \qquad A^\star = \mathcal{M} \cup \{\mathcal{I}\},$$

satisfies $\lambda_{\min}(\Sigma_{A^\star A^\star}) \geq \kappa_{\min} > 0$ and the method for constructing the Gram matrix is defined in assumption (A7).

(A5) (*Irrepresentability*) There exists $\eta > 0$ such that

$$\|\Sigma_{A^{\star c} A^\star} \Sigma_{A^\star A^\star}^{-1}\|_{2,\infty} \leq 1 - \eta.$$

(A6) (*Signal strength*) With group weights $w_g \in [1, C_w]$ and tuning parameter $\lambda_n \asymp \sigma\sqrt{\frac{\log G}{n}}$ (where $G$ is the number of candidate groups),

$$\min_{j \in \mathcal{M}} \|f_j\| \geq c_0 \lambda_n, \qquad \|f_\mathcal{I}\| \geq c_0 \lambda_n \;\; \text{if the interaction is present,}$$

for some $c_0 > 2/\eta$.

(A7) (*Finite orthonormal basis representation*) Each function $f_j$ and the interaction $f_\mathcal{I}$ is represented in an orthonormal basis expansion of finite dimension (at most quadratic order), with corresponding design blocks $\Phi_j$ and $\Phi_\mathcal{I}$.

Having specified the assumptions, we now turn to the proof. The role of (A7) is to provide a finite orthonormal basis representation of all effects, which allows us to formulate the regression problem as a finite-dimensional block group-lasso. Assumptions (A1)–(A6) then control the noise, dependence, eigenstructure, and signal strength needed to verify that the primal–dual witness construction recovers the correct support with probability tending to one.

By (A7), each main effect $f_j$ and the interaction $f_\mathcal{I}$ admits a finite-dimensional orthonormal basis representation, say

$$f_j(x_j) = \Phi_j(x_j)^\top \beta_j, \qquad f_\mathcal{I}(\mathbf{X}_\mathcal{I}) = \Phi_\mathcal{I}(\mathbf{X}_\mathcal{I})^\top \gamma,$$

where $\Phi_j \in \mathbb{R}^{n \times m_j}$ and $\Phi_\mathcal{I} \in \mathbb{R}^{n \times m_\mathcal{I}}$ collect the basis evaluations across $n$ samples. Stacking these blocks gives the design matrix

$$X = [X_1, \ldots, X_k, X_\mathcal{I}], \qquad X_j := \Phi_j, \quad X_\mathcal{I} := \Phi_\mathcal{I},$$

with block coefficient vector $\theta = (\beta_1, \ldots, \beta_k, \gamma)$. The true active set is $A^\star = \mathcal{M} \cup \{\mathcal{I} : f_\mathcal{I} \neq 0\}$ and the inactive set is $I^\star = \mathcal{G} \setminus A^\star$, where $\mathcal{G}$ denotes all candidate groups.

The SDAMI estimator solves the block group-lasso problem

$$\widehat{\theta} \in \arg\min_{\theta} \; \frac{1}{2n}\|y - X\theta\|_2^2 + \lambda_n \sum_{g \in \mathcal{G}} w_g \|\theta_g\|_2,$$

with tuning parameter $\lambda_n \asymp \sigma\sqrt{\frac{\log G}{n}}$ and group weights $w_g \in [1, C_w]$. The associated KKT conditions are

$$\frac{1}{n} X_g^\top (y - X\widehat{\theta}) = \lambda_n w_g \widehat{z}_g, \qquad \|\widehat{z}_g\|_2 \leq 1, \qquad \widehat{z}_g = \frac{\widehat{\theta}_g}{\|\widehat{\theta}_g\|_2} \;\; \text{if } \widehat{\theta}_g \neq 0.$$

Assumption (A1) ensures that the error vector $\varepsilon$ is sub-Gaussian. By a union bound over all blocks and coordinates, with probability $1 - o(1)$ the event

$$\max_{g \in \mathcal{G}} \frac{1}{n} \|X_g^\top \varepsilon\|_2 \leq \tfrac{1}{2} \lambda_n w_g$$

holds, providing high-probability control of noise terms in the KKT system. Assumptions (A2) and (A3) impose within-block orthonormality and block coherence, ensuring that $\Sigma = X^\top X / n$ has bounded eigenvalues and limited inter-block correlations. Assumption (A4) states a restricted eigenvalue condition, which guarantees that for any deviation vector $\Delta_{A^\star}$ supported on the active set,

$$\frac{1}{n} \|X_{A^\star} \Delta_{A^\star}\|_2^2 \geq \kappa_{\min} \|\Delta_{A^\star}\|_2^2.$$

Assumption (A5) provides the irrepresentability condition, ensuring that inactive blocks cannot mimic active ones in the dual constraints. Finally, assumption (A6) requires minimal signal strength $\|f_g\| \geq c_0 \lambda_n$ on all active blocks, so that true coefficients dominate the estimation error.

Under these conditions, the restricted problem on $A^\star$ yields an estimator $\widehat{\theta}_{A^\star}$ with error bound

$$\|\widehat{\theta}_{A^\star} - \theta_{A^\star}^\star\|_2 \leq \frac{3\lambda_n}{\kappa_{\min}} \Big( \sum_{g \in A^\star} w_g^2 \Big)^{1/2}.$$

Because $c_0 > 2/\eta$, this error is asymptotically smaller than the true signal size, ensuring $\widehat{\theta}_g \neq 0$ for all $g \in A^\star$. Thus, no active block is missed. For inactive groups, the dual feasibility condition requires $\frac{1}{n} \|X_g^\top (y - X_{A^\star} \widehat{\theta}_{A^\star})\|_2 < \lambda_n w_g$. The residual expands as $\widehat{r} = \varepsilon - X_{A^\star}(\widehat{\theta}_{A^\star} - \theta_{A^\star}^\star)$. The first term is controlled by (A1), while the second is bounded by (A3) and (A5) together with the error rate above. Consequently, inactive groups satisfy strict dual feasibility, forcing $\widehat{\theta}_g = 0$ for all $g \in I^\star$. This establishes absence of false positives.

For the interaction, if $f_\mathcal{I} = 0$, then $\mathcal{I} \in I^\star$ and the dual condition implies $\widehat{f}_\mathcal{I} = 0$. If $f_\mathcal{I} \neq 0$, then $\mathcal{I} \in A^\star$ and the signal strength bound ensures $\widehat{f}_\mathcal{I} \neq 0$. Combining all pieces, with probability tending to one we have

$$\{j : \widehat{f}_j \neq 0\} = \mathcal{M}, \qquad \widehat{f}_\mathcal{I} \neq 0 \Leftrightarrow f_\mathcal{I} \neq 0,$$

which proves the effect-level selection consistency of SDAMI as stated in Theorem 4.2.

## D    Conditions and Proof of Theorem 4.3

To ground the proof, we first specify the SDAMI function class and estimator used throughout.

**Model class of SDAMI.**    Let $A \subseteq \{1, \ldots, p\}$ index a subset of active main effects and interactions. For each main effect $j \in A_{\mathrm{main}}$ and interaction $\mathcal{I} \in A_{\mathrm{int}}$, let $\mathcal{N}_{L,W,B}$ denote the class of feedforward ReLU subnetworks of depth $L$ and maximal width $W$ whose parameters satisfy a norm constraint (e.g., path norm, spectral norm, or $\ell_2$ decay) bounded by $B$. For a growth schedule $(L_n, W_n, B_n)$, define the SDAMI sieve over $A$ by

$$\mathcal{F}_n^{\mathrm{SDAMI}}(A) = \left\{ f(x) = \sum_{j \in A_{\mathrm{main}}} g_j(x_j) \, + \, h_\mathcal{I}(x_\mathcal{I}) \, : \, g_j \in \mathcal{N}_{L_n, W_n, B_n}, \; h_\mathcal{I} \in \mathcal{N}_{L_n, W_n, B_n} \right\}.$$

Thus SDAMI is an additive model with interactions, where each component is realized by a subnetwork from $\mathcal{N}_{L_n, W_n, B_n}$ restricted to its own argument(s).

**Assumptions.**

(B1) *Sampling, noise, and approximation.* The data $(\mathbf{X}_i, Y_i)_{i=1}^n$ are i.i.d. from model (1) in the main content with $\epsilon_i$ satisfying $E[\epsilon_i] = 0$ and $\mathrm{Var}(\epsilon_i) = \sigma^2 < \infty$. The covariates $\mathbf{X}$ have either bounded support or sub-Gaussian tails, and the true regression function $f^\star \in L_2(P_\mathbf{X})$ lies in the $L_2(P_\mathbf{X})$-closure of the sieve

$$\bigcup_{n=1}^\infty \mathcal{F}_n^{\mathrm{SDAMI}}(A),$$

so that for any $\varepsilon > 0$ there exists $n$ and $f \in \mathcal{F}_n^{\mathrm{SDAMI}}(A)$ with $\|f - f^\star\|_{L_2(P_{\mathbf{X}})} \leq \varepsilon$.

(B2) *Effect-level selection consistency (SDAMI).* Let $A^\star$ be the true set of active main effects and interactions. Then $\mathbb{P}(\widehat{A}_n = A^\star) \to 1$.

(B3) *Approximation (DNN sieve over true inputs).* For the restricted DNN class $\mathcal{F}_n^{\mathrm{DNN}}(A^\star)$ with schedule $(L_n, W_n, B_n)$, the sieve approximation error vanishes:

$$\alpha_n := \inf_{f \in \mathcal{F}_n^{\mathrm{DNN}}(A^\star)} P\big[(f - f_{A^\star}^\star)^2\big] \longrightarrow 0.$$

(B4) *Empirical risk minimization up to tolerance.* The trained $\widehat{f}_n \in \mathcal{F}_n^{\mathrm{DNN}}(\widehat{A}_n)$ satisfies

$$P_n\big[(\widehat{f}_n - Y)^2\big] \leq \inf_{f \in \mathcal{F}_n^{\mathrm{DNN}}(\widehat{A}_n)} P_n\big[(f - Y)^2\big] + \delta_n, \qquad \delta_n \downarrow 0.$$

(B5) *Capacity control and uniform generalization.* The norm constraint $B_n$ (and/or width $W_n$) ensures a vanishing complexity for squared loss:

$$\mathfrak{R}_n(\mathcal{L}_n) = o(1), \qquad \mathcal{L}_n := \{(f - g)^2 : f \in \mathcal{F}_n^{\mathrm{DNN}}(A), \, g \in \mathcal{F}_n^{\mathrm{DNN}}(A), \, A \subseteq \{1, \ldots, p\}\},$$

so that

$$\sup_{h \in \mathcal{L}_n} \big|(P - P_n)h\big| = o_p(1).$$

(B6) *(Measurability and uniform $L_2$ envelope)* Each $f \in \mathcal{F}_n^{\mathrm{SDAMI}}(A)$ is measurable, and there exists a constant $M < \infty$ (independent of $n$, $A$, and $f$) such that

$$\sup_{A \subseteq [p]} \sup_{f \in \mathcal{F}_n^{\mathrm{SDAMI}}(A)} Pf^2 \leq M.$$

In particular, for the data–dependent active set $\widehat{A}_n$, the trained $\widehat{f}_n \in \mathcal{F}_n^{\mathrm{SDAMI}}(\widehat{A}_n)$ is measurable and satisfies $P\widehat{f}_n^2 \leq M$ almost surely. Hence $\{P\ell(\widehat{f}_n)\}_n$ is uniformly integrable.

With the SDAMI sieve $\mathcal{F}_n^{\mathrm{SDAMI}}(\widehat{A}_n)$ specified and assumptions (B1)–(B6) in place, we now prove Theorem 4.3 by analyzing the empirical minimizer within this class and translating vanishing risk into prediction convergence.

Let $P$ denote expectation with respect to $P_{\mathbf{X}}$ and $P_n$ the empirical average over the training inputs. Write the squared excess prediction loss as $\ell(f) := (f - f^\star)^2$. By the selection consistency of SDAMI (B2), $\mathbb{P}(\widehat{A}_n = A^\star) \to 1$, so it suffices to analyze $\widehat{f}_n \in \mathcal{F}_n^{\mathrm{SDAMI}}(A^\star)$ and the conclusions will then hold unconditionally. Using the empirical-to-population decomposition,

$$P\ell(\widehat{f}_n) = P_n\ell(\widehat{f}_n) + (P - P_n)\ell(\widehat{f}_n).$$

To control $P_n\ell(\widehat{f}_n)$, expand the empirical squared loss around $Y = f^\star + \epsilon$:

$$P_n\big[(\widehat{f}_n - Y)^2\big] = P_n\ell(\widehat{f}_n) + P_n[\epsilon^2] + 2P_n\big[(f^\star - \widehat{f}_n)\epsilon\big].$$

By the empirical optimality up to tolerance (B4), for any $f \in \mathcal{F}_n^{\mathrm{SDAMI}}(A^\star)$,

$$P_n\ell(\widehat{f}_n) \leq P_n\ell(f) + 2\Big|P_n\big[(f^\star - \widehat{f}_n)\epsilon\big]\Big| + 2\Big|P_n\big[(f^\star - f)\epsilon\big]\Big| + \delta_n.$$

The noise is centered with bounded conditional variance (B1) and the SDAMI sieve is capacity–controlled (B5), hence the stochastic inner products above are $o_p(1)$ uniformly over $f \in \mathcal{F}_n^{\mathrm{SDAMI}}(A^\star)$ by standard symmetrization/contraction bounds for squared loss. Taking the infimum over $f \in \mathcal{F}_n^{\mathrm{SDAMI}}(A^\star)$ yields

$$P_n\ell(\widehat{f}_n) \leq \inf_{f \in \mathcal{F}_n^{\mathrm{SDAMI}}(A^\star)} P_n\ell(f) + o_p(1) + \delta_n.$$

Adding and subtracting population risks and invoking the uniform generalization bound for squared loss from (D5),

$$P\ell(\widehat{f}_n) \leq \inf_{f \in \mathcal{F}_n^{\mathrm{SDAMI}}(A^\star)} P\ell(f) + o_p(1) + \delta_n.$$

| | Numerical Studies | Chip | Diabetes | V1 Cell |
|---|---|---|---|---|
| $\lambda_1$ | [0.01, 5) | [0.01, 1.5) | [1, 10) | [0.001, 0.03) |
| $\lambda_2$ | logspace[-3, 1) | logspace[-3, 1) | logspace[-3, 1) | logspace[-3, 1) |

| | Wine | Bikeshare | CA Housing |
|---|---|---|---|
| $\lambda_1$ | [0.1, 10) | [0.01, 2.5) | [0.001, 4.5) |
| $\lambda_2$ | logspace[-1, 1) | logspace[0.6, 1) | logspace[-1, 1) |

Table 5: Continuous Bandwidths for different task in the three-stage procedure.

By the approximation property of the SDAMI sieve on the true inputs (B3), the approximation error $\alpha_n := \inf_{f \in \mathcal{F}_n^{\mathrm{SDAMI}}(A^\star)} P\ell(f)$ satisfies $\alpha_n \to 0$; therefore

$$P\ell(\widehat{f}_n) \xrightarrow{p} 0. \tag{5}$$

To convert result (5) into prediction convergence, note the inequality

$$\mathbf{1}\Big\{\big|\widehat{f}_n(\mathbf{X}) - f^\star(\mathbf{X})\big| \geq \varepsilon\Big\} \ \leq\ \frac{\ell(\widehat{f}_n)(\mathbf{X})}{\varepsilon^2}, \qquad \varepsilon > 0.$$

Taking expectation over $\mathbf{X}$ and then over the training sample gives

$$\mathbb{P}\Big(\big|\widehat{f}_n(\mathbf{X}) - f^\star(\mathbf{X})\big| \geq \varepsilon\Big) \ \leq\ \frac{\mathbb{E}\big[P\ell(\widehat{f}_n)\big]}{\varepsilon^2}.$$

The sieve's norm constraints together with (B6) imply a square–integrable envelope on $\mathcal{F}_n^{\mathrm{SDAMI}}(A^\star)$, hence $\{P\ell(\widehat{f}_n)\}_n$ is uniformly integrable; combined with result (5) this yields $\mathbb{E}[P\ell(\widehat{f}_n)] \to 0$. Consequently,

$$\mathbb{P}\Big(\big|\widehat{f}_n(\mathbf{X}) - f^\star(\mathbf{X})\big| \geq \varepsilon\Big) \ \longrightarrow\ 0 \quad \text{for every fixed } \varepsilon > 0,$$

i.e., $\widehat{f}_n(\mathbf{X}) \xrightarrow{p} f^\star(\mathbf{X})$ at the design distribution $P_{\mathbf{X}}$. $\qquad\square$

# E    Supplementary Material for Hyperparameters Selection

In order to tune the hyperparameters, we performed a random stratified split of full training data into train set (80%), validation set (10%), and testing set (10%) for all datasets. For datasets we compile of small-sized with sparsity (Chips, Diabetes, V1-cell), and medium-sized (Wine Quality, Bikeshare, and California Housing), we do a 5-fold cross validation for 5 different test splits. Besides, we summarize the detail of cross validation on architecture selection, additional experiment results, and the visualization of either main effects or interactions effects from the numerical studies.

## E.1    SDAMIs and DNNs

Before building the neural network, the SDAMI's three-stage procedure requires careful tuning of regularization parameters. For the SpAM Screening, the $\lambda_1$ penalty is selected via Mallows $C_p$ where we set the basis dimension to 8. Subsequently, the $\lambda_2$ penalty is selected via 5-fold cross-validation with convergence tolerance is 0.0001. However, we have to design appropriate vector for $\lambda_1$ and use $C_p$ value as selection criteria to determine the optimal $\lambda_1$. We tune the penalty term in the three-stage procedure for each task in the continuous bandwidths and summarize in Table 5.

To determine the optimal neural network architecture for SDAMI and DNN baselines for numerical studies and small-sized dataset, we perform 5-fold cross-validation over three candidate configurations for each. Each configuration specifies the number and width of hidden layers in the subnetworks. We summarize the result of cross validation on configuration selection for SDAMI and DNN in Table 6, and the hyperparameter specification in Table 7.

| Method | SDAMI(1) | | SDAMI*(2) | | SDAMI(3) | | DNN(1) | | DNN*(2) | | DNN(3) | |
|--------|------|------|------|------|------|------|------|------|------|------|------|------|
| | MSE↓ | STD↓ | MSE↓ | STD↓ | MSE↓ | STD↓ | MSE↓ | STD↓ | MSE↓ | STD↓ | MSE↓ | STD↓ |
| Case 1 | 2.64 | 3.23 | **0.43** | 0.65 | 0.48 | 0.71 | 14.11 | 0.71 | 14.10 | 0.73 | 13.95 | 0.68 |
| Case 2 | 0.94 | 1.04 | 0.38 | 0.62 | **0.29** | 0.56 | 5.31 | 0.40 | 5.26 | 0.31 | 5.23 | 0.30 |
| Case 3 | 1.20 | 1.21 | 0.46 | 0.63 | **0.29** | 0.45 | 5.76 | 0.39 | 5.70 | 0.32 | 5.62 | 0.26 |
| Case 4 | 0.94 | 1.03 | 0.34 | 0.55 | **0.32** | 0.58 | 7.07 | 0.48 | 6.98 | 0.38 | 6.95 | 0.36 |
| Case 5 | 0.72 | 0.90 | **0.35** | 0.58 | 0.37 | 0.65 | 5.80 | 0.38 | 5.78 | 0.35 | 5.74 | 0.36 |
| Case 6 | 0.33 | 0.23 | **0.25** | 0.21 | 0.25 | 0.21 | 1.03 | 0.17 | 0.99 | 0.20 | 0.37 | 0.19 |

Table 6: (RMSE) Performance of SDAMs and DNNs with respect to different configuration when $n = 300$.

| Hyperparameter | numerical studies/ small-sized dataset | medium-sized dataset |
|----------------|----------------------------------------|----------------------|
| Architecture | [8, 6, 3], [15, 12, 10], [32,16,8] | [128, 64, 32, 16], [128, 64, 32], [64, 32, 16] |
| Batchsize | 16, 32, 64 | 1024, 2048 |
| Learning rate | 5e-2, 1e-2, 1e-3, 5e-3, | 5e-2, 1e-2, 1e-3, 5e-3, |
| Activation | ReLu | ReLu |
| Dropout | 0.0, 0.1 | 0.0, 0.1 |

Table 7: Model Specification for SDAMIs and DNNs

## E.2 FSPAM

We use fSpAM package (Ravikumar et al., 2009) and set the basis dimension as 8 with coordinate descent solver, and and best $\lambda$ penalty among $\{0.01, 0.05, 0.1, 0.5\}$ for 5 times and return the best model.

## E.3 LASSO AND LASSONET

We use LASSO package (Tibshirani, 1996) with default setting and best $\lambda$ penalty among $\{0.001, 0.01, 0.1, 1.0\}$ via 5-fold cross validation. As for LASSONET (Lemhadri et al., 2021), we consider the same architecture in Table 7 and best $\lambda$ penalty among $\{0.001, 0.01, 0.1\}$ for model comparison.

## E.4 NAM

We utilize NAM package (Agarwal et al., 2021) with number of embedded =32, number of hidden neuron =32, number of layers=3, and the learning rate=0.0005.

## E.5 GAMI-NET

We utilize the GAMI-NET PyTorch code (Yang et al., 2021). We set the interact number =10, subnetwork size of main effect= (20,), subnetwork size of interaction=(20, 20), learning rates=(0.001, 0.001, 0.0001), and loss threshold=0.01 and set early stop 100 rounds to ensure convergence.

## E.6 NODE-GAM AND NODE-GA$^2$M

We utilize the . We utilize the default hyperparameters from NODE-GAM PyTorch code (Chang et al., 2021), and set the number of trees to a large number 500, arch = GAM, learning rate = 0.01, warm-up = 100, and max epoch = 20000 to ensure it converges.

# F OPTIMAL HYPERPARAMETERS FOUND IN EACH DATASET

Here we report the best hyperparameters we find for 3 small-sized datasets and 4 medium-sized datasets in Table 8.

| Hyperparameter | numerical studies | Chip | Diabetes | V1 Cell | Wine | BikeShare | CA Housing |
|---|---|---|---|---|---|---|---|
| Architecture | [15, 12, 10] | [8, 6, 3] | [32, 16, 8] | [15, 12, 10] | [128, 64, 32, 16] | [128, 64, 32] | [128, 64, 32] |
| Batchsize | 32 | 64 | 64 | 32 | 2048 | 2048 | 2048 |
| Learning rate | 5e-2 | 1e-3 | 5e-2 | 5e-2 | 5e-2 | 1e-2 | 1e-3 |
| Activation | ReLu | ReLu | ReLu | ReLu | ReLu | ReLu | ReLu |
| Dropout | 0.0 | 0.0 | 0.0 | 0.0 | 0.0 | 0.0 | 0.0 |

Table 8: The best model specification for SDAMI architecture

|  | SDAMI | DNN | fSpAM | LASSO | NAM | GAMI-NET | NODE-GA$^2$M | NODE-GAM |
|---|---|---|---|---|---|---|---|---|
| Case 1 | $\mathbf{0.56}_{\pm 1.04}$ | $14.11_{\pm 0.71}$ | $5.57_{\pm 0.31}$ | $3.32_{\pm 0.24}$ | $10.10_{\pm 0.92}$ | $2.62_{\pm 0.63}$ | $2.66_{\pm 0.29}$ | $2.74_{\pm 0.50}$ |
| Case 2 | $\mathbf{0.30}_{\pm 0.57}$ | $5.31_{\pm 0.40}$ | $3.04_{\pm 0.16}$ | $2.54_{\pm 0.17}$ | $5.23_{\pm 0.43}$ | $2.26_{\pm 1.05}$ | $1.52_{\pm 0.70}$ | $1.97_{\pm 0.36}$ |
| Case 3 | $\mathbf{0.27}_{\pm 0.37}$ | $5.31_{\pm 0.40}$ | $3.04_{\pm 0.16}$ | $2.54_{\pm 0.17}$ | $4.01_{\pm 0.43}$ | $1.27_{\pm 0.52}$ | $0.71_{\pm 0.25}$ | $2.26_{\pm 0.45}$ |
| Case 4 | $\mathbf{0.24}_{\pm 0.41}$ | $7.07_{\pm 0.51}$ | $3.32_{\pm 0.17}$ | $2.80_{\pm 0.16}$ | $4.65_{\pm 0.59}$ | $1.55_{\pm 0.73}$ | $0.72_{\pm 0.35}$ | $2.29_{\pm 0.59}$ |
| Case 5 | $\mathbf{0.41}_{\pm 0.76}$ | $5.80_{\pm 0.38}$ | $3.45_{\pm 0.16}$ | $2.98_{\pm 0.20}$ | $3.84_{\pm 0.53}$ | $1.05_{\pm 0.61}$ | $0.57_{\pm 0.19}$ | $2.14_{\pm 0.44}$ |
| Case 6 | $\mathbf{0.35}_{\pm 0.21}$ | $1.03_{\pm 0.17}$ | $0.60_{\pm 0.03}$ | $0.43_{\pm 0.03}$ | $0.70_{\pm 0.06}$ | $0.29_{\pm 0.13}$ | $0.42_{\pm 0.03}$ | $0.45_{\pm 0.04}$ |

Table 9: (RMSE) The performance for 6 different case type when $n = 300$; $\downarrow$ means the lowest the better while $\uparrow$ means the highest the better.

# G  SUPPLEMENTARY MATERIAL FOR ADDITIONAL EXPERIMENT RESULTS

## G.1  COMPLETE COMPARISON FOR NUMERICAL STUDIES

The performance comparison among different machine learning model is demonstrated in Table 9 10, and the results of SDAMI$-p$ for both the numerical studies and real data analysis in Table 11 12. Also, Table 13 results for additional numerical experiments with different sample size and corresponding TPR/ FPR are demonstrated in the following block.

## G.2  VISUALIZATION FOR NUMERICAL STUDIES

In the section, we demonstrate the visualization of either main effects or interaction among each cases where the visualization result for Case 3 can be found in Figure 2. In Case 1 - 5, the SDAMI can capture both linearity and nonlinearity underlying the true model. In the interaction-existed cases, we can observes the SDAMI can still depict the response surface to approximate the underlying higher-order effects.

In this section, we demonstrate visualizations of the component functions representing either main effects or interactions across different cases. For Cases 1 through 5, SDAMI successfully captures both the linear and nonlinear structures underlying the true models. In cases involving interactions, we observe that SDAMI effectively depicts the response surfaces, accurately approximating the underlying higher-order effects. These visualizations provide valuable insights into the model's interpretability and can be found in detail in the Figure 4, 5.

# H  COMPUTATIONAL COMPLEXITY ANALYSIS

The proposed three-stage procedure achieves computational efficiency by progressively reducing the search space. The SpAM Screening fit $k$ univariate smooth functions via coordinate descent, with complexity $\mathcal{O}(p \cdot n \log(n))$, where $\log(n)$ reflects smoothing spline computations. Subsequently, the group Lasso decomposition expands each screened variable into an orthogonal basis of dimension $b$ and applies group Lasso with $e$ iterations, yielding complexity $\mathcal{O}(epbn)$. Lastly, the neural network fitting with each subnetworks with depth $L$ and width $W$ for approximate $E$ epochs. The total complexity is $\mathcal{O}(p \cdot n \log(n) + epnb + EnLW^2)$.

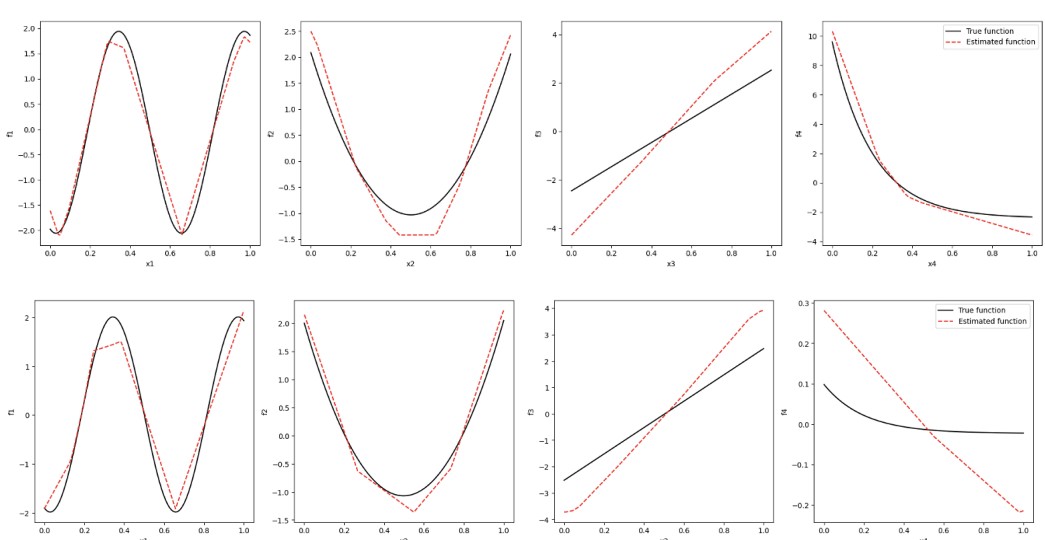

Figure 4: The estimated (red dashed lines) versus true additive component functions (solid black lines) for four main effects for (Upper panel) Case (1) and (Lower panel) Case (2).

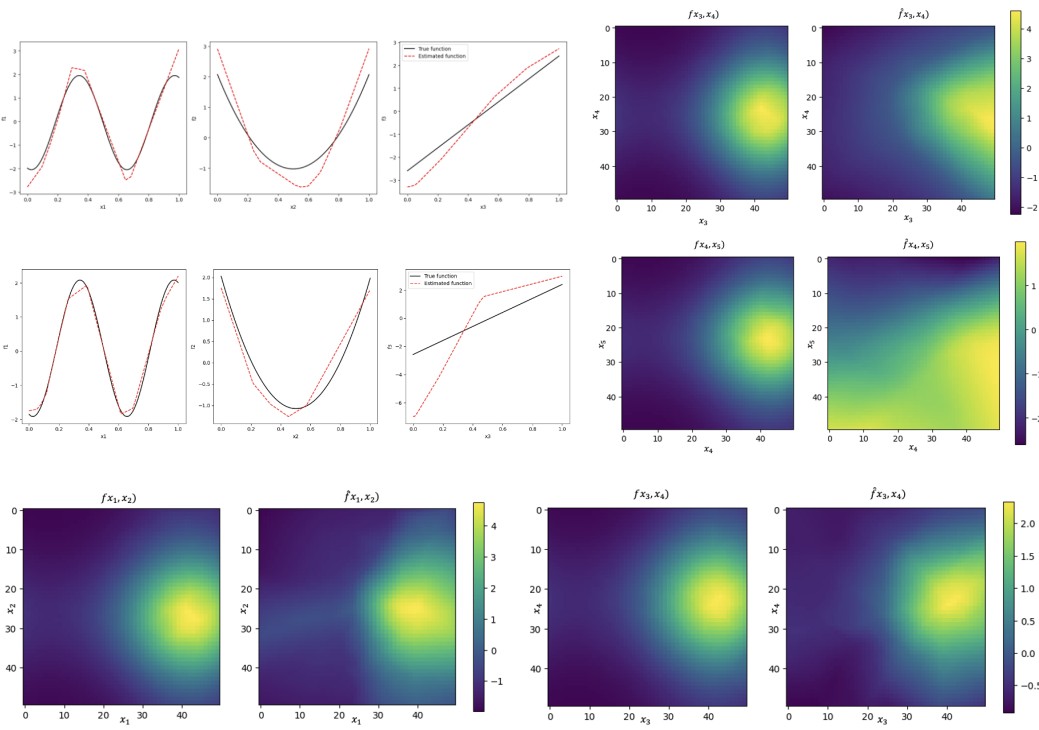

Figure 5: (Upper panel: Case (4); middle panel: Case (5)) The three figures on the left: Estimated (red dashed lines) versus true additive component functions (solid black lines) for three main effects; the two figures on the far right: the first shows the true response surface for interaction, and the second shows its estimated response surface. (Lower panel: Case (6)) The first and third shows the true response surface for interactions, and the second and fourth shows corresponding estimated response surface.

| | SDAMI | DNN | fSpAM | LASSO | NAM | GAMI-NET | NODE-GA$^2$M | NODE-GAM |
|---|---|---|---|---|---|---|---|---|
| Case 1 | **0.21**$_{\pm 0.12}$ | 13.89$_{\pm 0.82}$ | 5.43$_{\pm 0.28}$ | 3.04$_{\pm 0.17}$ | 5.60.$_{\pm 0.98}$ | 1.49$_{\pm 0.74}$ | 0.35$_{\pm 0.10}$ | 1.62$_{\pm 0.48}$ |
| Case 2 | **0.14**$_{\pm 0.04}$ | 5.33$_{\pm 0.35}$ | 2.98$_{\pm 0.14}$ | 2.40$_{\pm 0.11}$ | 1.71$_{\pm 0.30}$ | 0.73$_{\pm 0.36}$ | 0.25$_{\pm 0.07}$ | 1.20$_{\pm 0.34}$ |
| Case 3 | **0.17**$_{\pm 0.13}$ | 5.78$_{\pm 0.32}$ | 3.33$_{\pm 0.16}$ | 2.72$_{\pm 0.14}$ | 2.02$_{\pm 0.39}$ | 0.81$_{\pm 0.35}$ | 0.40$_{\pm 0.08}$ | 1.57$_{\pm 0.35}$ |
| Case 4 | **0.25**$_{\pm 0.53}$ | 7.14$_{\pm 0.50}$ | 3.24$_{\pm 0.15}$ | 2.61$_{\pm 0.13}$ | 2.54$_{\pm 0.41}$ | 0.93$_{\pm 0.35}$ | 0.39$_{\pm 0.06}$ | 1.49$_{\pm 0.35}$ |
| Case 5 | **0.26**$_{\pm 0.18}$ | 5.82$_{\pm 0.39}$ | 3.41$_{\pm 0.15}$ | 2.76$_{\pm 0.13}$ | 2.09$_{\pm 0.48}$ | 0.59$_{\pm 0.35}$ | 0.38$_{\pm 0.08}$ | 1.45$_{\pm 0.35}$ |
| Case 6 | **0.16**$_{\pm 0.12}$ | 1.06$_{\pm 0.13}$ | 0.59$_{\pm 0.03}$ | 0.39$_{\pm 0.02}$ | 0.52$_{\pm 0.04}$ | 0.16$_{\pm 0.05}$ | 0.37$_{\pm 0.03}$ | 0.37$_{\pm 0.03}$ |

Table 10: (RMSE) The performance for 6 different case type when $n = 450$; ↓ means the lowest the better while ↑ means the highest the better.

| | | Case 1 | Case 2 | Case 3 | Case 4 | Case 5 | Case 6 |
|---|---|---|---|---|---|---|---|
| | n = 150 | 0.68 $_{\pm 0.59}$ | 0.77 $_{\pm 0.41}$ | 0.70 $_{\pm 0.58}$ | 0.84 $_{\pm 0.27}$ | 0.85 $_{\pm 0.53}$ | 0.27 $_{\pm 0.25}$ |
| SDAMI$-p$ | n = 300 | 0.48 $_{\pm 0.71}$ | 0.29 $_{\pm 0.56}$ | 0.29 $_{\pm 0.45}$ | 0.32 $_{\pm 0.58}$ | 0.37 $_{\pm 0.65}$ | 0.25 $_{\pm 0.21}$ |
| | n = 450 | 0.23 $_{\pm 0.63}$ | 0.21 $_{\pm 0.46}$ | 0.28 $_{\pm 0.37}$ | 0.17 $_{\pm 0.21}$ | 0.22 $_{\pm 0.19}$ | 0.14 $_{\pm 0.18}$ |

Table 11: (RMSE) The performance for 6 different case type for SDAMI$-p$; ↓ means the lowest the better while ↑ means the highest the better.

In high-dimensional settings where the full pairwise interaction space is $\binom{k}{2}$, SDAMI's three-stage decomposition yields substantial computational savings: by focusing only on variable screened. The computational cost is summarized in Table 14.

# I    REAL DATA ANALYSIS

This section illustrates the additional experiment on two real datasets with redundant features including the parameter settings and corresponding explanation on the visualization. Besides, we also use data without redundant features such as wine quality, bike share, and California housing. The description is summarized in Table 15.

## I.1    SURROGATE MODELING OF PRODUCT LIFETIME MODELING

This subsection showcases the application of SDAMI in evaluating prediction performance, positioning it as an effective surrogate technique— a key approach in the field of computer experiments (Santner et al., 2019; Wu & Hamada, 2011). Surrogate modeling serve as statistical approximations of computationally intensive simulations, facilitating the efficient study of complex system dynamics.

We illustrate this with the analysis of electronic device lifetimes, which can fail due to mechanisms such as front-end fate oxide breakdown (FEOL TDDB) (Yang et al., 2017). This failure occurs when traps accumulate in the gate oxide layer from electrical and thermal stress during operation, eventually creating conductive paths leading to device malfunction. The lifetime distribution for these components is captured by the following function, as characterized in prior work (Hsu et al., 2020):

$$S(t) = \exp\left(-\left(\frac{t}{A_{\text{FEOL}}(\text{WL})^{-\frac{1}{\beta}} e^{-\frac{1}{\beta}} V^{a+bT} \exp\left(\frac{cT+d}{T^2}\right) s^{-1}}\right)^{\beta}\right), \tag{6}$$

where the inputs include process-dependent constants $A_{\text{FEOL}}, a, b, c, d$, voltage $V$ and temperature $T$, width $W$ and length $L$ of the device, the probability of stress $s$, and shape parameter $\beta$ describing failure progression over time.

| | Chip | Diabetes | V1 Cell | Wine | BikeShare | CA Housing |
|---|---|---|---|---|---|---|
| SDAMI$-p$ | 0.236 | 52.87 | 0.372 | 0.692 | 55.91 | 0.508 |

Table 12: (RMSE) The performance for 7 different real dataset for SDAMI$-p$; ↓ means the lowest the better while ↑ means the highest the better.

| Method | SDAMI | | LASSONET | | SODA | |
|---|---|---|---|---|---|---|
| | TPR↑ | FPR↓ | TPR↑ | FPR↓ | TPR↑ | FPR↓ |
| Case 1 | **1.0000** (-) | $1.1 \times 10^{-5}$ (-) | 0.6100 (0.1241) | 0.0129 (0.0091) | 0.03 (0.0964) | $5 \times 10^{-4}$ ($2 \times 10^{-4}$) |
| Case 2 | **1.0000** (-) | $1.1 \times 10^{-5}$ (-) | 0.4550 (0.1083) | 0.0168 (0.0083) | 0.02 (0.0685) | $4 \times 10^{-4}$ ($2 \times 10^{-4}$) |
| Case 3 | **0.7500** (-) | $10^{-4}$ ($10^{-5}$) | 0.0200 (0.0980) | 0.0451 (0.0418) | 0.015 (0.06) | $6 \times 10^{-4}$ ($4 \times 10^{-4}$ |
| Case 4 | **0.7600** (0.0490) | $10^{-4}$ ($10^{-5}$) | 0.0100 (0.0700) | 0.0367 (0.0176) | 0.025 (0.0758) | $5 \times 10^{-4}$ ($2 \times 10^{-4}$) |
| Case 5 | **0.7525** (0.0249) | $10^{-4}$ ($10^{-5}$) | 0.0100 (0.0700) | 0.0390 (0.0183) | 0.03 (0.0821) | $5 \times 10^{-4}$ ($2 \times 10^{-4}$) |
| Case 6 | **0.6100** (0.0436) | $10^{-4}$ ($10^{-5}$) | 0.0200 (0.0980) | 0.0519 (0.0658) | - (-) | $5 \times 10^{-4}$ ($2 \times 10^{-4}$) |

Table 13: Mean (standard deviation) of TPR and FPR over 100 simulations from SDAMI, LAS-SONET, SODA when $n = 300$ where $(-)$ indicates value $< 1e^{-5}$.

Table 14: Computational Cost Comparison Across Methods and Datasets. Times reported in seconds, averaged over 10 runs for simulation and real dataset on GPU: Tesla V100-SXM2-32GB.

| Method | Runtime (seconds) | | | |
|---|---|---|---|---|
| | **Simulation** $(n = 150, p = 150)$ | **Wine** $(n = 4892, p = 12)$ | **BikeShare** $(n = 17379, p = 12)$ | **CA Housing** $(n = 20640, p = 8)$ |
| SDAMI | 6.04 | 16.40 | 77.54 | 78.62 |
| DNN | 0.42 | 58.84 | 58.6 | 7.02 |
| fSpAM | 0.001 | 0.004 | 0.008 | 0.005 |
| LASSO | 0.01 | 0.01 | 0.02 | 0.01 |
| LASSONET | 43.72 | 168.04 | 363.88 | 391.01 |
| NAM | 7.04 | 19.78 | 65.31 | 61.24 |
| GAMI-Net | 38.22 | 23.96 | 55.67 | 40.62 |
| NODE-GAM | 130.55 | 458.33 | 1199.69 | 1871.75 |
| NODE-GA$^2$M | 185.75 | 448.15 | 1858.62 | 1864.26 |

Although simulating such experiments is straightforward, accurately extracting main and higher-order effects under data sparsity requires sophisticated and interpretable modeling. To that end, we employ the *MaxPro design* (Joseph et al., 2015) to generate space-filling experiments spanning all input factors, with details in Table 16. The dataset includes 100 observations with 9 covariates, augmented by 21 irrelevant noise features randomly sampled uniformly within $[0, 1)$ to test model sparsistency and interaction detection. The log-transformation of the true model is given by

$$\log(\eta) = \log(\mathrm{A}_{FEOL}) - \frac{1}{\beta} \log(\mathrm{WL}) - \frac{1}{\beta} + (a + bT)\log(V) + \left(\frac{cT + d}{T^2}\right) - \log(s), \quad (7)$$

where $s$ is constant and $\eta$ corresponds to a 63% failure quantile from the generalized Wei-bull model (6). This representation admits an additive decomposition involving univariate and bi-variate functions 7,

$$y = \alpha + \sum_i f_i(x_i) + \sum_{i \neq j} f_{ij}(x_i, x_j) + \cdots + \epsilon.$$

allowing comprehensive identification of relevant main and interaction effects. Table 4 presents the comparative performance of various techniques, including SDAMI, NAM, GAMI-Net, NODE-GAM, NODE-GA$^2$M, DNN, LASSO, LASSONET, and fSpAM, demonstrating SDAMI's prominence in recovering complex dependency structures in sparse, high-dimensional settings.

Given the visualization of effects from Figure 6, we can observe that the contribution of main effect is relatively weak. Besides, the interaction have obvious effect on response. To be more specific, when

| Dataset | Source | Samples | Features | Description |
|---|---|---|---|---|
| Chip | (Hsu et al., 2020) | 100 | 9 baseline + 21 noise | MOSFET device lifetime |
| Diabetes | scikit-learn | 200 | 10 baseline + 40 noise | Serum measurements |
| V1 fMRI | Kay et al. (2008) | 300 | 1800 Gabor | Primary visual cortex responses |
| Wine Quality | UCI ML Repository | 4898 | 11 | Wine quality |
| BikeShare | UCI ML Repository | 17379 | 12 | Capital Bikeshare hourly rental counts |
| California Housing | scikit-learn | 20640 | 8 | Median house values |

Table 15: Table of Real Datasets with Sources

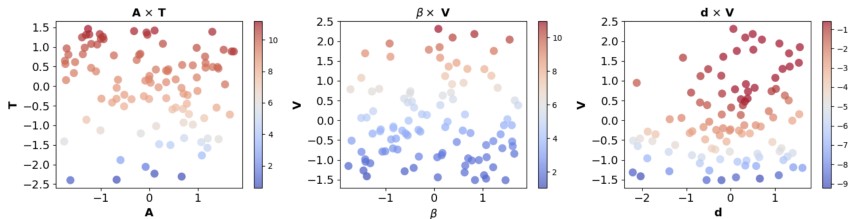

Figure 6: The shape plots of 3 Interactions of SDAMI trained on Chip Dataset.

$(A, T)$, $(d, V)$ and $(\beta, V)$ goes up, the response will increase. These phenomenon is predictable because in Equation 7, the higher-order effects are dominant over main effects but the main effects still exist due to its marginal effect on the response.

| Parameter | Lower | Upper |
|---|---|---|
| a | $-81.9$ | $-74.1$ |
| b | $7.69 \times 10^{-2}$ | $8.51 \times 10^{-2}$ |
| c | $8.37 \times 10^{3}$ | $9.25 \times 10^{3}$ |
| d | $-8.14 \times 10^{5}$ | $-7.33 \times 10^{5}$ |
| $\beta$ | $1.476$ | $1.804$ |
| V | $1.2$ | $1.3$ |
| T | $120$ | $180$ |
| WL | $4 \times 10^{-4}$ | $6 \times 10^{-4}$ |
| $A_{\text{FEOL}}$ | $4.75 \times 10^{-7}$ | $5.25 \times 10^{-7}$ |
| s | $1$ | $1$ |

Table 16: Parameter table for generating space-filling experiment on MOSFET device

### I.2 DIABETES RESPONSE PREDICTION

For this analysis, we utilize the well-known diabetes dataset from the scikit-learn library, which contains 442 observations and ten baseline covariates. These features capture key demographic and physiological measurements, such age (in years), sex (0: female, 1: male), body mass index (BMI), mean arterial blood pressure, and six standardized blood serum variables known to be relevant for diabetes progression. The target variable is a quantitative measure of disease progression observed one year after baseline, making the dataset suitable for regression modeling and biomarker analysis.

To thoroughly evaluate sparse additive modeling methods under high-dimensional constraints, we purposefully restrict the sample size to $n = 200$ and augment the original dataset with $40$ synthetic covariates, each drawn independently from a uniform distribution on the interval $[0, 1)$ distribution. These additional features are explicitly designed to act as non-informative noise, challenging each model's ability to discern relevant predictors. Thus, the expanded dataset includes 50 covariates in total, with the genuine signal confined to the original ten baseline measurements. Standard preprocessing, including normalization and scaling of all features, is performed to ensure comparability and numerical robustness in downstream modeling. This controlled, high-dimensional experimental setup provides a rigorous testbed for assessing the sensitivity and variable selection performance of SpAM, and other advanced machine learning algorithms in biomedical contexts.

Visualization of the estimated effects in Figure 7 reveals several interpretable patterns. There are one main effect and three interactions term selected by SDAMI. First, the log of Serum Triglycerides Level ($s5$) has a positive association with diabetes disease progression. The research shows that among patients with type 2 diabetes, those with elevated $s5$ had significantly worse glycemic control even when treated with insulin (Zheng et al., 2019). Second, higher disease progression when (high BMI and elevated $s5$) and (high BMI and elevated blood pressure) are present simultaneously. These combination defines the beginning stages of Cardiovascular-Kidney-Metabolic syndrome, which dramatically accelerates diabetes complications. The observed relationships align well with clinical expectations and domain knowledge.

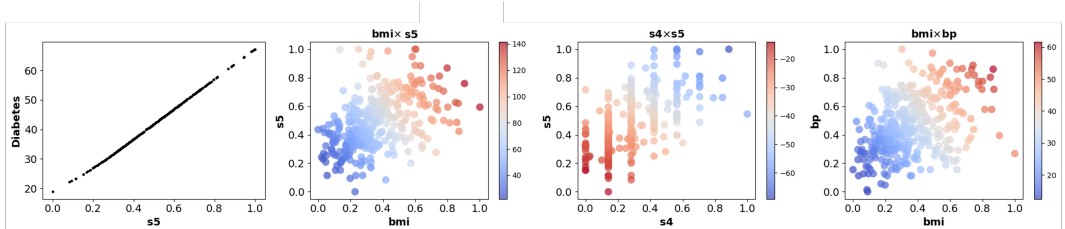

Figure 7: (Diabetes Dataset) The first figures on the left: Predicted marginal response of target with respect to main effect feature; the three figures on the right: the shape plots of 3 Interactions of SDAMI trained on Diabetes Dataset.

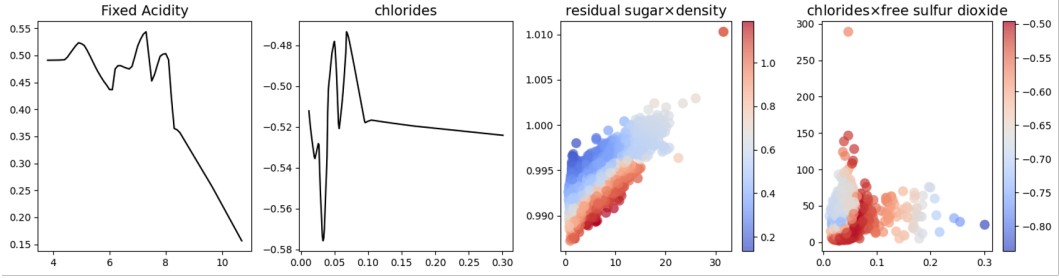

Figure 8: The shape plots of selected main effects and interaction of SDAMI trained on Wine Dataset.

Table 4 summarizes model performance for diabetes response prediction. SDAMI with interaction modeling consistently outperform alternative machine learning methods, offering superior predictive accuracy alongside enhanced interpretability thanks to its explicit feature selection and effect visualization capabilities.

We select the relative important main effects and interaction term for Wine, Bikeshares, and California housing where the shape plot demonstrate the relationship between features and the targeted response. The shape plots are provided in Figure 8, 9, and 10.

### I.3    HUMAN PRIMARY VISUAL CORTEX DATASET

In Figure 11, these images are first passed through localized, orientation- and phase-sensitive Gabor filters to mimic simple-cell receptive fields; outputs then undergo nonlinear transforms to produce single-cell responses. Complex cells are formed by pooling quadrature-phase pairs (square–sum–nonlinearity), yielding phase-invariant responses. Subsequently, in Figure 12, these complex cells will be fed into Group Lasso, and be identified as single input (main effect) or composite input (interaction)

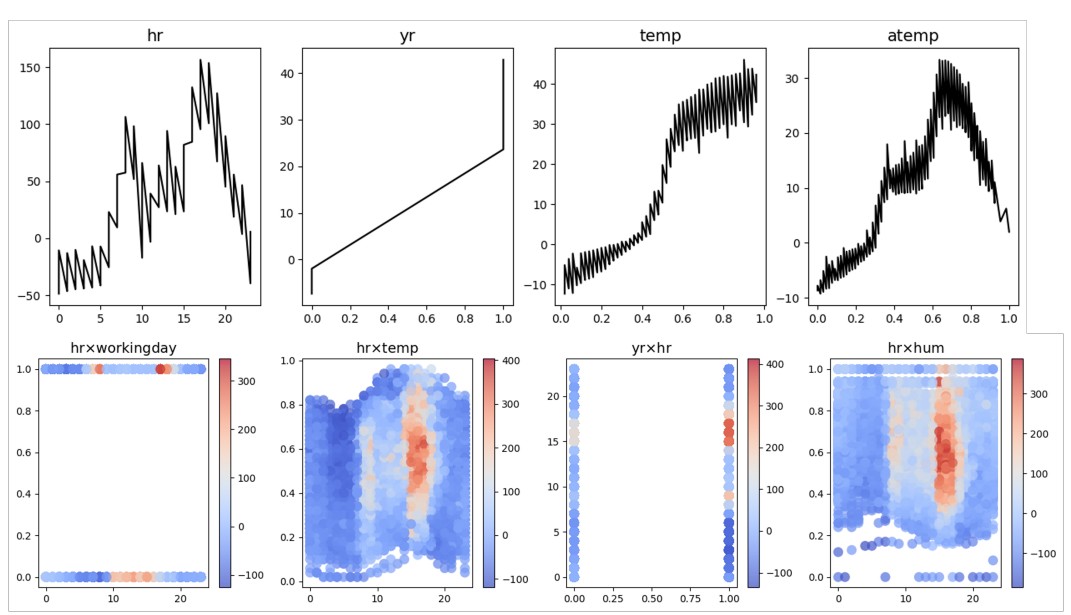

Figure 9: The shape plots of selected main effects and interaction of SDAMI trained on Bikeshares Dataset.

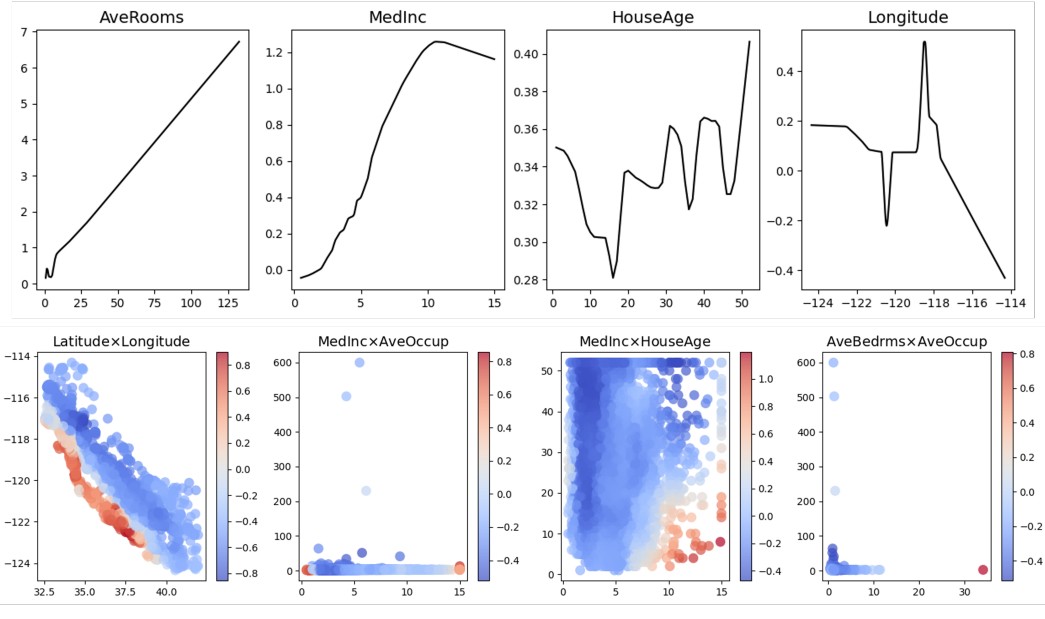

Figure 10: The shape plots of selected main effects and interaction of SDAMI trained on California Housing Dataset.

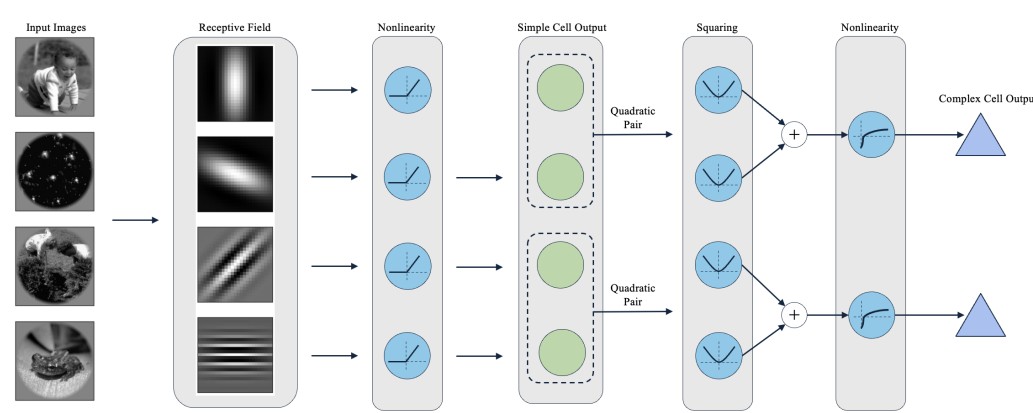

Figure 11: The formation of complex cells arises from nonlinear activation of quadratic pairs of simple cells generated by Gabor-wavelet filters applied to the input.

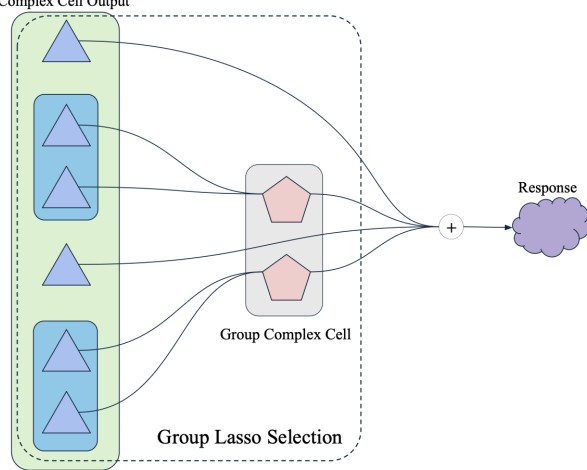

Figure 12: The formation of response arises from complex cells and group complex cells selected by group lasso applied to the input.

