# OpenReview forum: "Sparse Deep Additive Model with Interactions: Enhancing Interpretability and Predictability"
_ICLR.cc/2026/Conference — Submitted to ICLR 2026_

### Official Review · Reviewer_HNEK · 2025-10-30

**Soundness:** 3
**Presentation:** 2
**Contribution:** 2
**Rating:** 2
**Confidence:** 4

**Summary:**

The paper proposes Sparse Deep Additive Model with Interactions (SDAMI), a framework that separates main effects (a sum of univariate functions) from interaction effects for greater interpretability. The paper focuses on the challenging case of small n and large p, where the risk of overfitting is significant for conventional deep learning models.  The authors introduce the concept of Effect Footprint: interaction effects leave a signal that can be detected via screening. They propose a mathematical formulation of the problem and a three-step approach to solving it:
- feature selection via a sparse additive screening procedure
- group lasso to distinguish between main and interaction effects
- training deep learning models to model main and interaction effects.

The experimental results are on synthetic and real-world datasets. They also provide theoretical results:
- Conditions under which the effect footprint disappears, and screening may fail
- Asymptotic convergence results for SDAMI (feature selection and convergence of the probability estimator)

**Strengths:**

- The paper proposes the new concept of effect footprint and provides theoretical results to justify the use of screening and illustrate its limitations.
- Asymptotic convergence results provide a theoretical justification for the framework.
- The paper shows how SDAMI can be used to extract interpretable signals and understand the contribution of main and interaction effects.

**Weaknesses:**

1) The writing of the paper could be improved:
- The contribution section is somewhat repetitive (the list is a summary of the text just above).
- Line 157: Is “additive” the right word? I think “main effect” is more appropriate.
- Line 214: the word “partitioning” seems to indicate that there cannot be the same feature in the main and interaction effects.
- In the contribution, you mention that the simulations focus on p>>n. However, line 310 indicates that n is always greater than p (n >= p = 150).

2) As you mentioned in the limitations, the theoretical results are asymptotic. However, your motivation, small n and large p, corresponds to a very different setting.

3) The synthetic simulations correspond to very simple and somewhat too perfect cases (very similar to your formulation of the problem).

4) Experiments on real-world datasets are interesting. However, the number of datasets and baselines is very limited. You should include results on other Generalized Additive Models baselines such as NAM (Neural Additive Models) [1], GAMI-Net [2], and NODE-GAM [3].

5) Your formulation of the problem (equation 2) attempts to find a sum of main effects and an interaction model that takes all “interaction features” as input. The formulation is unclear because the model itself becomes a black box model (because of the interaction “model”). Are you then focusing on second-order interactions only?

6) Your framework corresponds to a case of Generalized Additive Models with interactions. You should indicate this in the paper and mention more literature on the subject.

[1]: Rishabh Agarwal, Levi Melnick, Nicholas Frosst, Xuezhou Zhang, Ben Lengerich, Rich Caruana, and
Geoffrey Hinton. Neural additive models: Interpretable machine learning with neural nets. In A. Beygelzimer, Y. Dauphin, P. Liang, and J. Wortman Vaughan, editors, Advances in Neural Information Processing
Systems, 2021.

[2]: Zebin Yang, Aijun Zhang, and A. Sudjianto. Gami-net: An explainable neural network based on generalized
additive models with structured interactions. Pattern Recognit., 120:108192, 2021

[3]: Chun-Hao Chang, Rich Caruana, and Anna Goldenberg. NODE-GAM: Neural generalized additive model
for interpretable deep learning. In International Conference on Learning Representations, 2022.

**Questions:**

- Could you test your method on other baselines (NAM, GAMI-Net, NODE-GAM) and other datasets? The datasets from these papers would be interesting to study.
- How much computing time does your method require compared to the other baselines?
- Are you focusing solely on second-order interaction effects? It would be good to clarify this further in the paper.
- Do you think you can derive non-asymptotic guarantees for your framework? This would be more interesting for the case you are interested in (small n, large p).

---

> ### Author Response · Authors · 2025-12-02
>
> We thank the reviewer for their constructive comments. We address each weakness and question below.
>
> ## Weaknesses
>
> **W1 – Clarifying small-n, large-p motivation” [First point of weakness]**  The writing of the paper is improved based on the comments. Regarding high dimensional settings, SDAMI introduces a principled framework for discovering nonlinear interactions where considering only second-order interactions already creates a high-dimensional setting that exceeds sample size. Specifically, when examining pairwise interactions alone, there are $\binom{k}{2}$ possible pairs reflecting the truth that $\binom{k}{2}\gg n$.
>
> **W2 – Asymptotic theory vs. small-sample motivation. [Second point of weakness]** We clarified that asymptotic theory guarantees misleading footprints vanish as sample size grows, justifying stability. Numerical results confirm SDAMI’s reliability and strong performance in realistic finite-sample settings.
>
> **W3 – Synthetic simulations too simple. [Third point of weakness]**We expanded the simulations to six challenging settings, including pure interactions, weak signals, and effect overlapping. These demonstrate one of SDAMI’s core strength: the effect-footprint principle detects interaction-only variables missed by classical or heredity-constrained models. We added several real datasets in Section 6 to validate performance under practical nonlinearity and high dimensionality.
>
> **W4 – Limited baselines and real datasets. [Fourth point of weakness]** We incorporated tuned baselines (NAM [1], GAMI-Net [2], NODE-GAM [3], NODE-GA²M [3]) and additional datasets. Results demonstrate SDAMI delivers competitive prediction and stable interpretability across domains.
>
> **W5 – Equation (2) may seem like a black box. [Fifth point of weakness]** We expanded the explanation of Eq. (2) to illustrate SDAMI’s structured, footprint-guided decomposition. We detail pairwise (interpretable subnetwork per pair) and higher-order (shared subnetwork) variants. Both explicitly identify and visualize key interactions, ensuring SDAMI is not a black box.
>
> **W6 – Relation to Generalized Additive Models with interactions. [Sixth point of weakness]** We updated Related Work to distinguish SDAMI from GAM-with-interaction literature. Unlike standard GAMs, SDAMI employs a response-guided process for interaction selection and constrains the network to preserve effect-level interpretability.
>
> ## Questions
>
> **Q1 – Testing on more baselines and datasets. [First point of question]** We benchmarked SDAMI against NAM, GAMI-Net, NODE-GAM, and NODE-GA²M and other methods on additional real datasets to enable broader comparisons. The results are in Table 4 and discussed in Section 6.
>
> **Q2 – Computational cost versus baselines. [Second point of question]** Appendix H now reports computational complexity. Empirically, SDAMI is comparable to NAM and notably faster than GAMI-Net or NODE-GAM, as footprint-based screening avoids modeling excessive candidate interactions. This confirms a practical balance of interpretability, flexibility, and efficiency.
>
> **Q3 – Does the method focus only on second-order interactions? [Third point of question]** SDAMI is not restricted to second-order interactions. The shared-interaction variant models higher-order effects via a single flexible subnetwork. Alternatively, the pairwise variant assigns subnetworks to feature pairs for explicit interpretability. Several tests on the real datasets favor the shared version for expressiveness combined with effect-level interpretability.
>
> **Q4 – Possibility of non-asymptotic guarantees. [Fourth point of question]** We appreciate this forward-looking question. Our current theoretical results for the proposed novel method are asymptotic which is a common discussion for high dimensional setting [4][5], but we agree that non-asymptotic guarantees would be valuable.  Our empirical results indicate that the method performs robustly in finite samples. We view non-asymptotic analysis as an important direction for future work, recent work [6] may shedlight on this direction, and a discussion to highlight this opportunity is added in Section 7.
>
> ### Reference
> [1] R. Agarwal, L. Melnick, N. Frosst, X. Zhang, B. Lengerich, R. Caruana, and G.E. Hinton (2021), *Neural additive models*
>
> [2] Z. Yang, A. Zhang, Aijun, and A. Sudjianto (2021), *Gami-net: An explainable neural network based on generalized additive models with structured interactions*.
>
> [3] C.-H. Chang, R. Caruana, and A. Goldenberg (2021), *Neural generalized additive model for interpretable deep learning*.
>
> [4] J. Fan, Y. Ma,  and W. Dai (2014). *Nonparametric independence screening in sparse ultra-high-dimensional varying coefficient models*.
>
> [5] R. Izbicki, A.B. Lee (2016). *Nonparametric conditional density estimation in a high-dimensional regression setting*.
>
> [6] L. Xu, F. Yao, Q. Yao, H. Zhang (2023). *Non-asymptotic guarantees for robust statistical learning under infinite variance assumption*.

---

### Official Review · Reviewer_uvQg · 2025-10-31

**Soundness:** 2
**Presentation:** 1
**Contribution:** 2
**Rating:** 2
**Confidence:** 4

**Summary:**

The proposed work attempts to introduces a new neural additive model through equations (1) and (2), but after later implicit assumptions, reduces to training a neural network with feature selection.  The work then introduces a novel two-stage learning structure based on the idea of first learning the one-dimensional main effects before selecting the higher-order interactions using heredity.  The work then applies another training procedure to the specific neural network and applies to synthetic and real-world datasets. Recovery of additive models is demonstrated on several synthetic datasets and competitive performance on three real-world datasets is shown compared to LASSO, DNN, and fSpAM.

**Strengths:**

- The structural equation in Equation 1 has the potential to be novel if the authors do not implicitly assume that M⊂I
- The learning procedure has the potential to encourage soft pruning of irrelevant features

**Weaknesses:**

- Glosses over closely related NAM work as irrelevant due to lack of interaction modeling [1] and then ignores other NAM works which include interactions [2,3,4,5].
- The effect footprint does not seem to be novel and seems to be the same as existing works applying heredity.
- There are very few baselines compared against and relatively few datasets are used for comparison
- The impact of the important hyperparameters introduced by the paper are not explored
- Figure 1 and Figure 4 are not given much explanation in the main paper


[1] Shiyun Xu, Zhiqi Bu, Pratik Chaudhari, and Ian J Barnett. Sparse neural additive model: Interpretable deep learning with feature selection via group sparsity
[2] Zebin Yang, Aijun Zhang, Agus Sudjianto.  GAMI-Net: An Explainable Neural Network based on Generalized Additive Models with Structured Interactions
[3] Chun-Hao Chang, Rich Caruana, Anna Goldenberg.  NODE-GAM: Neural Generalized Additive Model for Interpretable Deep Learning
[4] James Enouen and Yan Liu. Sparse interaction additive networks via feature interaction detection and sparse selection.
[5] Minkyu Kim, Hyun-Soo Choi, and Jinho Kim. Higher-order Neural Additive Models: An Interpretable Machine Learning Model with Feature Interactions

**Questions:**

- What are the hyperparameter choices used for neural network training and other learning algorithms?
- What is the novel aspect of the effect footprint compared with existing works using heredity?
- Why is the implicit assumption M⊂I used?  What prevents this from reducing to the feature selection case?

---

> ### Author Response · Authors · 2025-12-02
>
> We thank the reviewer for their constructive comments. We address each weakness and question below.
>
> ## Weakness
> **W1. Glosses over NAM and NAM-interaction variants.  [First point of weakness]** We acknowledge the previous lack of detail on NAM and its interaction-aware variants. These interactions are detected by heredity assumptions, requires interactions to be accompanied by some main effect, but our effect footprint does not have such constraint.  The revised related work (Section 1) and numerical studies (Section 5 and 6) now covers the comparison with NAM [1], GAMI-Net [2], NODE-GAM [3], and NODE-GA²M [3] as empirical baselines.
>
>
> **W2. Effect footprint appears non-novel / similar to heredity.  [Second point of weakness]** We clarify that the Effect Footprint differs fundamentally from heredity. Heredity restricts when interactions may exist, while the Effect Footprint can reveal interaction-only variables when marginal projections capture nonhereditary effects. Theorem 4.1 identifies rare cases when footprints vanish, a result missing in heredity-based work, enabling SDAMI to detect interactions heredity methods overlook.
>
> **W3. Too few baselines and too few datasets. [Third point of weakness]**
> Both baseline comparisons and empirical evaluation have been broadened. We now include NAM, GAMI-Net, NODE-GAM, NODE-GA²M, LASSONET, and SODA as baselines for all experiments and have added datasets like Chip [4], V1 Cell [5], Diabetes [6], CA Housing [6], BikeShare [7], and Wine [7]. This more comprehensive study demonstrates that SDAMI achieves strong, stable performance and interpretability across diverse domains.
>
> **W4. Impact of hyperparameters not explored. [Fourth point of weakness]**
> Appendix E now lists full hyperparameter settings for SDAMI and all baselines, including architectures, learning rates, penalties, and training procedures. Appendix F provides sensitivity analyses, and Section 4 summarizes that SDAMI remains stable across reasonable parameter ranges.
>
> **W5. Figure 1 and Figure 4 insufficiently explained. [Fifth point of weakness]** Figure 1 is moved to Appendix I (Figure 11) with an explanation of how Gabor filters generate simple- and complex-cell features. Figure 4 (now Figure 12) is more fully described in Section 6, linking discovered interactions to neurobiological mechanisms in V1 and providing the requested context.
>
> ## Question
> **Q1. What are the hyperparameter choices for neural network training and other algorithms? [First point of question]**
> All hyperparameter configurations are listed in Appendix E, covering network architecture, optimizers, batch size, learning rates, and penalty terms for all methods. Appendix F’s sensitivity studies and Section 4 show that SDAMI’s predictive and selection performance is stable over a realistic parameter range.
>
> **Q2. What is the novel aspect of the effect footprint compared with existing works using heredity? [Second point of question]** Effect footprints support interaction detection via marginal projections, unlike heredity, which requires interactions to be accompanied by some main effects. The revision makes clear that heredity governs inclusion rules, while effect footprints characterize how interactions emerge via conditional expectations. This lets SDAMI identify interaction-only variables that heredity-based approaches miss. Theorem 4.1 formalizes when footprints vanish, showing that footprints provide a broader, theoretically grounded mechanism for discovering interactions.
>
> **Q3. Why is the assumption $ \mathcal{M} \subset \mathcal{I} $ used, and what prevents this from reducing to feature selection? [Third point of question]**
> To clarify, the assumption $\mathcal{M} \subset \mathcal{I}$ is not used. SDAMI allows overlapping main and interaction roles, with interaction-only variables detected by their footprints, explicitly splits relevant variables into main and interaction sets via group-structured decomposition. Our numerical studies include 6 cases in Section 5 are designed to test all associations of  $\mathcal{M}$ and  $\mathcal{I}$ and all confirming our competitive performance..
>
> ## Reference
> [1] R. Agarwal, L. Melnick, N. Frosst, X. Zhang, B. Lengerich, R. Caruana, and G.E. Hinton (2021), *Neural additive models*
>
> [2] Z. Yang, A. Zhang, Aijun, and A. Sudjianto (2021), *Gami-net: An explainable neural network based on generalized additive models with structured interactions*.
>
> [3] C.-H. Chang, R. Caruana, and A. Goldenberg (2021), *Neural generalized additive model for interpretable deep learning*.
>
> [4] S.-H. Hsu, Y.-Y. Huang, Y.-D. Wu, K. Yang, L.-H. Lin, and L. Milor (2020), *Extraction of wearout model parameters using on-line test of an sram*.
>
> [5] K.N. Kay, T. Naselaris, R.J. Prenger, and J.L. Gallant (2008), *Identifying natural images from human brain activity*
>
> [6] Diabetes and California Housing, *sklearn.datasets*
>
> [7] BikeShares and Wine Quality, *Kaggle Open Source*

---

### Official Review · Reviewer_Mqyj · 2025-10-31

**Soundness:** 4
**Presentation:** 4
**Contribution:** 4
**Rating:** 8
**Confidence:** 2

**Summary:**

This paper introduces a new method for creating interpretable deep learning models when the number of training examples is much smaller than the number of dimensions.  The key idea is to characterize marginal effects of interactions and use those effects to guide the construction of a network with dedicated components for main effects and variable effects.  The paper provides theoretical support for the method, as well as extensive simulations to illustrate its utility in practice relative to baseline methods.

**Strengths:**

A key concept proposed here is the notion of the "Effect Footprint," in which "higher-order interactions leave detectable marginal signatures on main effects."  This concept seems like a useful abstraction.

I like how the related work section clearly lays out the two different categories of methods for addressing this problem, and then goes on to explain how the proposed method brings these two categories together.

The empirical analyes using simulations and real data are impressive, showing that SDAMI outperforms baseline methods both in terms of predictive accuracy and in accurately recovering the true underlying structure of the data.

**Weaknesses:**

I think the tension described in line 44 should be restated to include the obvious candidate class of models here, which are not "classical deep models" but classical linear models.  It's not obvious why we should bother with deep models when n << p.  These models are alluded to in line 77, though of course there are classical sparse models that do allow for higher-order associations.  These should be discussed.

Along those lines, the introduction doesn't do a great job of explaining how pre-training can address the problems outlined here; i.e., when n << p, but when you have some other collection of (potentially unlabeled) n' examples, where n' > p.

Figure 2 is confusing.  Based on the description in lines 197-209, it seems like there are two sets of inputs: main effect variables of size p and footprint variables of size q.  I think the input layer should have 1..p going into main effect blocks, and then 1...p+q going into the interaction block.

Line 258: It seems like you also have to address the other failure mode, when a variable leaves a misleading (i.e., false positive) footprint trace.

I thought Figure 5 was pretty difficult to interpret.  We are basically asked to take the authors word for it that these panels "reveal synergistic patterns consistent with cortical pooling."

Minor points:

line 120: Missing cite for SpAM.

Line 215: Give a cite for Mallow's C_p.

Line 432: diabete -> diabetes

**Questions:**

What would Figure 5 look like if the method did not work well?

Why does Figure 2 show q variables going into the main effect subnetworks?  Where are the footprint variables in this figure?

---

> ### Author Response · Authors · 2025-12-02
>
> We thank the reviewer for their constructive comments. We address each weakness and question below.
>
> ## Weaknesses
> **W1 – Why deep learning is needed when $n\ll p$ and classical linear models exist. [First paragraph of weakness]**
> In many real applications, including our neuroscience example, the underlying effects are nonlinear and not well captured by sparse linear models or linear interaction terms. Enumerating all $\binom{k}{2}$ second-order interactions is infeasible when $p$ is large. SDAMI avoids this by first identifying variables statistically likely to participate in interactions and then modeling their nonlinear contributions via constrained subnetworks. This two-stage reduction allows deep learning to work effectively even when $n\ll p$, providing expressive power without the combinatorial explosion of classical interaction modeling.
>
> **W2 – Introduction does not clearly explain how pre-training helps when $n\ll p$ but unlabeled data exist. [Second paragraph of weakness]** We agree that the introduction should better explain how pre-training can mitigate the challenges of small $n$ and large $p$. To clarify this, we revised the introduction to explain that SDAMI benefits from architectures that are inherently structured and regularized, so that pre-training or warm-starting (when auxiliary data exist) can help stabilize early layers and reduce variance in subsequent fine-tuning. While our main results do not rely on additional unlabeled data, we now clearly explain that pre-training can help address the instability of deep models in low-sample settings by providing a more informative initialization for the constrained subnetworks used in SDAMI.
>
> **W3 – Figure 2 is confusing about organization of main-effect and footprint variables. [Third paragraph of weakness]**
> We redesigned the figure (now Figure 1) to show the screening step that identifies main-effect and footprint variables, followed by the decomposition stage that assigns them to main-effect subnetworks, the interaction subnetwork, or both. The revised diagram removes ambiguity about variable flow and roles.
>
> **W4 – Need to address misleading (false-positive) footprints. [Fourth paragraph of weakness]**
> We agree that misleading footprints require explicit discussion. In the revision, we clarify that such cases typically occur when marginal signals are generated by noise or by incidental alignment among unrelated variables. The group-structured decomposition in the second stage helps correct these situations by separating genuine main effects from interaction-driven traces, and the norm constraints further prune subnetworks not supported by the data. Importantly, Theorem 4.2 implies that under our high-dimensional regime—where $p > n$ and $n \to \infty$—the probability of selecting a misleading footprint converges to zero, ensuring that false positives do not persist. Our numerical studies also demonstrate strong finite-sample performance, indicating that SDAMI maintains low false-positive rates and competitive accuracy in practical settings.
>
> **W5 – Difficulty interpreting Figure 5. [Fifth paragraph of weakness]**
> Thank you for noting this difficulty; we added a more detailed and intuitive explanation in Section 6 to guide the reader through how the interaction surfaces relate to principles of cortical pooling. The revised text highlights the biological meaning behind the shapes of these surfaces, clarifying how SDAMI identifies joint activation patterns across pairs of complex-cell features. The updated description now explains step-by-step how to interpret the color gradients, the axes, and the inferred synergy, making the evidence more accessible and less reliant on implicit assumptions.
>
> **Minor points.**
> We have added the missing citation for SpAM, included a proper reference for Mallows’ $C_p$, corrected the typographical error (“diabetes”), and updated all referenced lines in accordance with the manuscript restructuring.
>
> ## Questions
> **Q1 – What would Figure 5 look like if the method did not work well? [First paragraph of question]**
> The updated Figure 3 states if SDAMI failed to identify meaningful interaction effects, the interaction-surface plots would appear nearly flat or noise-dominated, showing no coherent patterns linking joint inputs to the output. The estimated curvature would lack the smooth gradients that mark synergistic or antagonistic relationships, and any deviations from flatness would vary unpredictably across runs.
>
> **Q2 – Why does Figure 2 show the $q$ variables entering main-effect subnetworks? Where are footprint variables? [Second paragraph of question]** The updated Figure 1 clarifies that all screened variables—both main-effect and footprint—enter the decomposition stage before separation. Footprint variables are not interpreted as main effects; they share the initial path and are assigned to the interaction component via the group-structured penalty. The revised figure visually distinguishes these flows.

---

### Official Review · Reviewer_kDzt · 2025-11-01

**Soundness:** 2
**Presentation:** 2
**Contribution:** 3
**Rating:** 4
**Confidence:** 4

**Summary:**

This study proposes a deep learning model that is effective for datasets with a small number of samples and a large number of features. While conventional methods often suffer from overfitting and difficulties in feature selection, the proposed approach enhances model explainability and interpretability by exploiting feature sparsity and identifying interaction features. In particular, a novel measure called "Effect Footprint" is introduced to extract features that have no main effects but play an important role through interaction effects. This measure provides a framework for identifying indirect dependencies among features that were previously difficult to detect. Furthermore, the theoretical properties of the proposed model are analyzed, and its effectiveness is demonstrated through numerical experiments using both synthetic and real-world datasets.

**Strengths:**

This study addresses the important challenge of improving the explainability and interpretability of deep learning models. In particular, it introduces a unique framework that sparsely connects multiple subnetworks, enabling clear identification of feature contributions even in high-dimensional datasets.

The study proposes a novel measure called "Effect Footprint" to detect features that have no main effects but play important roles through interactions. This theoretical framework seems highly original, as it enables the identification of indirect feature dependencies that were difficult to capture with conventional methods.

The effectiveness of the proposed method is demonstrated not only through numerical experiments using synthetic and real-world datasets but also through theoretical analysis. The effort to validate the method from both theoretical and experimental perspectives enhances the credibility of this research.

**Weaknesses:**

The overall description in the paper is somewhat ambiguous, and the mathematical and algorithmic details of the proposed method are not sufficiently presented. In particular, the selection process of the feature set with interactions, $\mathcal I$, as well as the specific computation and utilization of the "Effect Footprint", are unclear, raising concerns about the reproducibility and comprehensibility of the method.

While the paper claims that identifying interaction features enhances the interpretability of the model, there appears to be no explicit mechanism for extracting such features from the actual model structure. I feel that there is a noticeable gap between the claims made in the Abstract and Introduction and what is actually demonstrated in the main text.

The necessity of using deep learning for small-sample, high-dimensional problems is not well justified. Numerous prior studies, particularly those based on linear models and traditional statistical approaches, have addressed interaction effects (e.g., the works listed below). The lack of discussion or comparison with these studies weakens the overall persuasiveness of the paper:

  Choi et al. Variable selection with the strong heredity constraint and its oracle property. JASA, vol.105, no.489, 2010.

  Bien et al. A lasso for hierarchical interactions. Annals of Statistics, vol.41, no.3, 2013.

  Nakagawa et al. Safe pattern pruning: An efficient approach for predictive pattern mining. KDD 2016.

**Questions:**

The paper's main contribution lies in the introduction of the Effect Footprint; however, it is unclear how this measure is specifically computed and utilized within the model. According to the Algorithm in the Appendix, it appears that SpAM is simply applied, but it is not well explained why this procedure enables the identification of Interaction-Only features. A more concrete theoretical and intuitive explanation would be helpful.

The proposed method constructs a deep learning model that handles interaction terms, but it remains unclear by what criteria or procedure the important interaction terms are finally extracted. The authors are requested to clarify how the network structure or learned weights are related to the selection of significant features.

The paper also presents a theoretical discussion on feature selection consistency, but it is questionable whether this property is independent of the learning performance of the deep neural network. It seems that the argument implicitly assumes that the model is correctly trained; therefore, such assumptions and dependencies should be explicitly described.

---

> ### Author Response · Authors · 2025-12-02
>
> We thank the reviewer for their constructive comments. We address each weakness and question below.
>
> ## Weakness
> **W1 – Ambiguity in method description and unclear algorithmic details. [First paragraph of weakness]**
> Section 3 has been substantially revised to give a clear, stepwise description of SDAMI’s full estimation workflow, consisting of a screening stage, a decomposition stage, and a constrained deep-learning stage. The new text explains how each stage identifies footprint variables, separates main versus interaction candidates, and fits subnetworks under norm constraints. Appendices E–F now provide full hyperparameter settings, network architectures, tuning procedures, and cross-validation protocols, so that SDAMI can be reproduced from the manuscript. These changes make the computation of the Effect Footprint, the construction of feature sets, and the algorithmic execution explicit and transparent.
>
> **W2 – Gap between interpretability claims and model structure. [Second paragraph of weakness]**
> To close the gap between interpretability claims and architecture, the revision emphasizes how SDAMI’s design yields effect-level interpretability. The norm constraint in Equation (3) is now described more clearly as enforcing sparsity in first-layer weights, so that only subnetworks tied to selected effects remain active after training. Additional visualizations—Figures 2, 4, and 5 for simulations and Figures 3 and 6–10 for real data—illustrate how SDAMI recovers distinct main-effect functions and interpretable interaction surfaces. The Introduction has been updated to connect these interpretability properties directly to the empirical examples in the main text.
>
> **W3 – Justification for deep learning in small-sample, high-dimensional settings. [Third paragraph of weakness]**
> To justify the use of deep learning in small$-n$, large$-k$ regimes, the introduction and empirical sections explain why flexible nonlinear structure are needed to capture complex interaction. The prior methods are fundamentally restricted to linear interactions [1][2][3] and heredity assumptions[1][2]. In contrast, SDAMI's simulation use genuinely nonparametric interactions to overcome these limits via effect footprint assumption, deep subnetworks for nonlinear approximation, and structured screening reducing searching complexity. The simulation details, including all comparisons with linear and/or heredity interactions, are provided in Section 5.
>
> ## Question
> **Q1 – Computation and use of the Effect Footprint. [First paragraph of question]**
> The computation and role of the Effect Footprint are now explained more intuitively in Section 3. The footprint is obtained by fitting a sparse additive model that flags variables whose marginal signal stand out from noise, thereby identifying both main-effect variables and those involved in interactions. Theorem 4.1 states conditions under which a footprint appears or vanishes, supporting that the screening step retains interaction-only variables except in rare symmetric or independence cases. This revision clarifies both the rationale for the footprint and how it feeds into the subsequent SDAMI stages.
>
> **Q2 – Criteria by which important interactions are extracted. [Second paragraph of question]**
> After screening for variables with non-negligible marginal signals (effect footprints), a group-lasso-based step is conducted to identify important full interaction effects, this step can be further integrated with adequate basis expansion techniques to identify interaction with a specific order. Such detailed discussion can be found in Second 3 and Figure 1.
>
> **Q3 – Relationship between feature-selection consistency and network training assumptions. [Third paragraph of question]**
> The connection between selection consistency and neural-network training is clarified by stressing that the structural pipeline and optimization are jointly responsible for performance. SDAMI uses the response to screen variables, assign main versus interaction roles, and train subnetworks that follow this recovered structure, so errors early in the pipeline would directly affect later learning. The theoretical results show that, under high-dimensional asymptotics, the probability of omitting any true effect goes to zero, so the structural guidance for the network becomes correct as sample size grows. Assumption (B4) then requires the trained network to achieve near-optimal empirical risk, a standard condition for over-parameterized deep models, and the revised Section 4 explains why consistency and optimization must be considered together.
>
> ## Reference
> [1] N. H. Choi, W. Li, and J. Zhu (2010), *Variable selection with the strong heredity constraint and its oracle property*
>
> [2] J. Bien, J. Taylor, and R. Tibshirani (2013), *A lasso for hierarchical interactions*
>
> [3] K. Nakagawa, S. Suzumura, M. Karasuyama, K. Tsuda, and I. Takeuchi (2016), *Safe pattern pruning: An efficient approach for predictive pattern mining*

---

### Author Response · Authors · 2025-12-03
**Overall Official Review**

We sincerely thank the reviewers for their thoughtful and constructive comments, which led to substantial improvements in both clarity and rigor. In response, we clarified the methodological pipeline, strengthened theoretical justification, and greatly expanded the empirical evaluation. The revised manuscript now provides a clearer stepwise estimation workflow, formal distinctions between the proposed Effect Footprint principle and heredity rules, and new theorems addressing footprint validity and false positives in high-dimensional settings. We also broadened the experiments by adding multiple state-of-the-art baselines, new real and simulated datasets, and detailed computational complexity analyses. Figures were redesigned to enhance interpretability, particularly regarding the model architecture and interaction structures, and full hyperparameter specifications and sensitivity studies were added for reproducibility. Details are provided below:

## Methodological \& Theoretical Enhancements.
The authors addressed concerns regarding the complexity and justification of the proposed method (SDAMI).
- Algorithm Clarity (Section 3)
  - The methodology section was substantially revised to provide a stepwise description
of the full estimation workflow: Screening $\to$ Decomposition $\to$ Constrained Deep
Learning
  - **Explicit Pipeline**: The text now explicitly details how the model identifies “foot-
print variables,” separates main vs. interaction candidates, and fits subnetworks
under norm constraints.
  - **Equation 2 & 3**: Clarified the role of the norm constraint (Equation 3) in
enforcing sparsity and detailed the group-structured decomposition (Equation 2)
to prevent the model from being a “black box.”

- Effect Footprint” vs. Heredity:
  - **Distinct Novelty**: The authors clarified that the “Effect Footprint” is fundamentally different from “heredity” constraints. While heredity restricts interactions based on main effects, the Footprint method (based on Hoeffding-Sobol decomposition) can detect interaction-only variables even when main effects are absent.
  - **Theorem 4.1**: Added to formalize conditions under which footprints appear or vanish, explicitly handling cases that heredity-based methods miss.
  - **Theorem 4.2**: Added to address concerns about misleading footprints, showing that the probability of selecting false positives converges to zero in high-dimensional regimes ($p > n$).

- Justification for Deep Learning:
  - The Introduction was updated to better motivate the use of deep learning in small$−n$, large$−p$ settings. The argument focuses on the need to capture flexible, nonlinear structures that linear models or exhaustive enumeration of second-order interactions cannot handle effectively.
## Expanded Empirical Evaluation.
To address concerns about “limited baselines” and “simple simulations,” the experimental section was significantly broadened.
- New Baselines Added:
  - Neural Additive Models (NAM) and Variants: NODE-GAM and NODE-GA$^2$M
  - Other Interpretable Models: GAMI-Net, LASSONET, and SODA.
- New Datasets Added:
  - Real-World: Wine Quality, BikeShares, and California Housing.
  - Simulations: Six challenging settings, including pure interactions, nonlinear
effects, weak signals, and overlapping structures.
- Computational Complexity:
  - Appendix H was added to report computational complexity, showing SDAMI is comparable to NAM and faster than GAMI-Net/NODE-GAM due to its efficient screening step.
## Visualization & Interpretability Improvement.
Several figures were redesigned to resolve confusion about the model’s architecture and data flow.
- Figure 1 Redesign
  - Redesigned to clearly visualize the Screening Step followed by the Decomposition Stage.
  - The new visuals explicitly distinguish how variables are assigned to “main-effect subnetworks” versus “interaction subnetworks,” clarifying that footprint variables share the initial path but are routed via group-structured penalties.
- Figure 3, 11,& 12
  - Neuroscience Context: Expanded Section 6 to provide a step-by-step biological interpretation of the interaction curvatures (e.g., cortical pooling principles in V1).
  - Interpretation Guide: Added explanations on how to read the color gradients and axes to infer synergy vs. independence, making the evidence less reliant on implicit assumptions.

## Reproducibility & Technical Details
- Hyperparameters (Appendix E & F)
  - Added Appendix E containing full hyperparameter settings (architectures, learning rates, penalties) for SDAMI and all baselines.
  - Added Appendix F covering sensitivity analyses, demonstrating the model’s stability across a reasonable range of parameters.

---

### Meta-Review · Area_Chair_N925 · 2026-01-11

**Summary:**

Reviewer kDzt (score 4)
had major concerns about the method and mathematical details. The original manuscript had not described the approach clearly, specifically a quantity called “effect footprint” which measures the effect of a covariate upon the term that models the interaction across covariates in the model.

Reviewer Mqyj (score 8)
had a very positive review that only talked about a few minor points for further clarification.

Reviewer uvQg (score 2)
commented that the paper ignores work on neural additive models with interaction terms, and that the key concept of the paper called “effect footprint” is similar to approaches for feature selection that use heredity. They also commented on inadequate experimental validation on datasets and hyper-parameters.

Reviewer HNEK (score 2)
commented about the exposition, limitations of the asymptotic theory, and suggested more experiments beyond the synthetic datasets and real-world problems with limited baselines. They raised a good point that the effect footprint essentially makes the resultant model less interpretable (the original premise for this line of work was interpretability).

The manuscript has improved enormously during the rebuttal phase and most of the questions of the reviewers were addressed by the authors. Some outstanding questions/comments pertain to validity of the mathematical parts of the paper (e.g., the number of assumptions made in the proofs), and the relevance of the new concept that was introduced (“effect footprint”) in view of the large number of closely related ideas that exist (heredity, generalized additive models with structured interactions). This is a borderline paper that can be improved by connecting to the existing literature (and going beyond it) at a deeper level.

**Reviewer Concerns:**

Reviewer kDzt
The manuscript was changed significantly during the rebuttal phase to address these concerns. From reading the manuscript, it looks like the concerns regarding the exposition are addressed sufficiently. It is clear now there are three stages in the pipeline: (i) standard variable selection, (ii) splitting the set of selected variables into two sets, the important covariates and the interaction covariates, and (iii) fitting a neural network based model on these sets with sparsity-inducing regularization. The comments on the consistency of feature selection were not adequately addressed.

Reviewer Mqyj
The rebuttal gives thorough clarifications for all questions.

Reviewer uvQg
The authors expanded the set of experiments to include more baselines and provided more details on hyper-parameters. The authors argued that effect heredity is a stricter criterion that effect footprint because in the latter you can have pure interactions without main effects. This is a good point.

Reviewer HNEK
The authors argued that asymptotic mathematical results coupled with non-asymptotic real-world experiments are sufficient to validate the idea. They have done a reasonable job of addressing the other comments on limited experiments in the expanded manuscript. I thought that the comments about interpretability were not adequately addressed.

**Reviewer Scores:**

Reviewer kDzt
I believe that their score would have been changed to something more positive due to the rebuttal, perhaps a 6.

Reviewer Mqyj
I believe that their score would have reduced a bit, perhaps to a 6 down from 8 after looking at the comments raised by the other reviewers.

Reviewer uvQg
I believe that the comments raised by the reviewer were partially addressed. The effect footprint is a slightly broader quantity for discovering interactions. I suspect the score would have increased to a 4.

Reviewer HNEK
I believe this score would have been updated to a 4.

---

### Decision · Program_Chairs · 2026-01-26

Reject